# TPI-VA: THIRD-PARTY INTERRUPTION-AWARE VOICE ASSISTANT

## ABSTRACT

While recent progress in Spoken Language Models (SLMs) has enabled increasingly natural voice-based interactions, they remain vulnerable to third-party interruptions (TPI). To address this challenge, we present a holistic framework for building and evaluating TPI-aware voice assistants. We first introduce `TPI-Train`, a large-scale dataset of 80K instances spanning 26 realistic interruption scenarios. For evaluation, we introduce `TPI-Bench`, which includes `TPI-Test` for measuring response strategies under interruptions and `Janus-Test` for probing whether models can distinguish true multi-speaker utterances from acoustically single-speaker yet textually misleading speech. To ensure reproducible and interpretable assessments, we also design two complementary metrics: Response Strategy Following (RSF) and Overall Helpfulness (OH). Experiments demonstrate that models fine-tuned with our approach achieve robust performance on `TPI-Bench` while preserving general dialogue capabilities on VoiceBench, effectively avoiding reliance on textual shortcuts. Human evaluations further confirm that both our dataset and trained models align with human preferences, establishing the first comprehensive solution for TPI-aware voice assistants. Our dataset and the automation pipeline of the framework will be publicly available[1].

## 1 INTRODUCTION

Recent Spoken Language Models (SLMs) (Wu et al., 2025; Stacey et al., 2024a; Kim et al., 2024; Xu et al., 2025a) have significantly advanced the capabilities of voice assistants (VAs), enabling increasingly natural and human-like conversations. These models excel in dyadic interactions, adeptly handling complex queries and maintaining conversational flow with a single speaker. However, this proficiency is largely confined to dyadic interactions, as current models struggle to navigate the complexities of multi-party social contexts (Wang et al., 2025a). A practically significant and plausible failure case arises in scenarios involving third-party interruptions (TPI), where VAs often misinterpret a multi-speaker dialogue as a single, continuous utterance from the primary user. For instance, if a speaker asks, "Should we order the new pasta?" and a third-party interjects with, "No, let's just get the usual," an ordinary VA might process the entire sequence as a self-repair (Levelt, 1983) utterance from the initial speaker—a common phenomenon in voice assistant interactions (Goel et al., 2023; Stacey et al., 2024b; Liu et al., 2024). This erroneous concatenation of user input leads to nonsensical or inappropriate responses as described in Figure 1 and Appendix A, degrading the user experience, eroding trust, and ultimately discouraging further engagement with the voice assistant (Baughan et al., 2023).

We hypothesize that this shortcoming (Wang et al., 2025a) does not merely indicate deficiencies in dialogue-level reasoning but stem from a more fundamental limitation: a lack of sensitivity to acoustic cues. To bridge this gap, we propose a transition from dyadic language modeling to interruption-aware modeling. Building on this motivation, we argue that effective handling of TPI requires two essential abilities: (1) *Discerning Speaker Interruption*—the ability to robustly detect interruptions, which in turn enables consistent performance under single-speaker conditions, and (2) *Situation-Discriminative Response*—generating contextually appropriate replies in TPI situations, adapting strategies such as addressing or disregarding interruptions depending on user preferences.

---

[1]Demo samples: https://tpi-va.github.io/

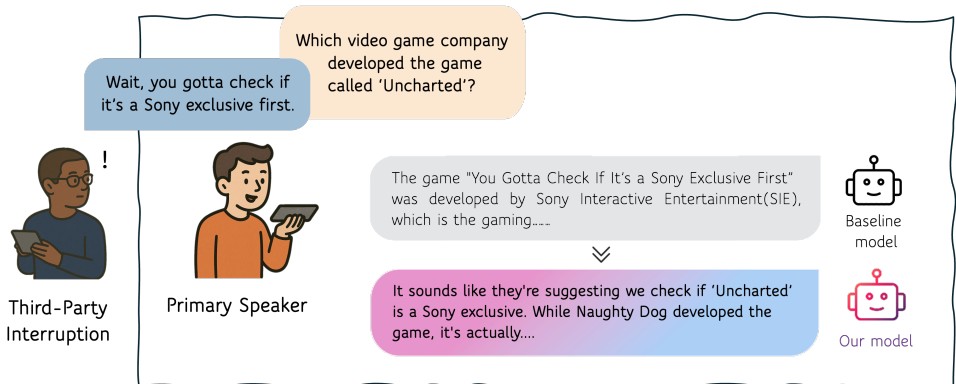

Figure 1: **Example of a TPI query sampled from our TPI-Corpus.** The spoken language model mistakes the third-party interruption for a continuous utterance from the primary speaker, while our model correctly identifies the interruption and responds in a TPI-aware manner. Failure cases are described in Appendix A.

Achieving the *situation-discriminative response*, however, presents a nuanced challenge: there is no universal one-size-fits-all "correct" response to interruptions (Xie et al., 2022; Cao et al., 2025). The decision of when, and how a VA should address a third-party intervention depends on subjective user preferences and the situational context (Tabassum et al., 2020). Thus, TPI handling cannot be solved by a rigid solution but instead demands a flexible framework that supports customed response strategies. To this end, we propose a comprehensive framework that unifies the entire workflow: from specifying response strategies, to constructing strategy-conditioned datasets, and to conducting corresponding evaluations. In addition, we instantiate a reference answer strategy within our framework and demonstrate, via human evaluations, that models trained under this strategy produce effective and natural responses under TPI situations.

As a foundation for this framework, we introduce `TPI-Corpus`, a large-scale dataset of 84K samples spanning 26 interruption scenarios, derived from extensions of canonical interruption taxonomies (Yang et al., 2022; Murata, 1994; Goldberg, 1990) and specifically adapted to third-party contexts. We partition this corpus into `TPI-Train` for building TPI-aware models and `TPI-Bench` for rigorous evaluation. `TPI-Bench` comprises two complementary benchmarks: `TPI-Test`, which evaluates models' ability to produce **situation-discriminative responses** under genuine interruptions, and `Janus-Test`, which probes whether models can **discern speaker interruption** from acoustically single-speaker yet textually misleading speech. To ensure reliable and interpretable assessment, we further propose two LLM-based metrics: Response Strategy Following (RSF) and Overall Helpfulness (OH).

Our experiments reveal that existing open-source SLMs struggle to handle third-party interruptions. Naively fine-tuning on `TPI-Train` alone imparts interruption-handling ability but also induces an over-reliance on semantic shortcuts. To address this, we design a composite training approach where each data source serves a distinct role: `TPI-Train` provides interruption-specific supervision, single-speaker dialogues preserve core conversational competence, and a small set of carefully constructed hard negatives enforces reliance on acoustic evidence over textual cues. This strategy produces a balanced and robust TPI-aware model without sacrificing general abilities. Its ability to discriminate input scenarios is supported by well-separated embedding representations and further validated by human evaluations, which confirm that both our dataset and trained model generate responses aligned with user preferences. Together, these results underscore the practical value of our framework for building TPI-aware voice assistants.

Our contributions are summarized as follows:

- We define *TPI-awareness* and establish the first comprehensive framework for achieving it, centered on `TPI-Corpus`, which is divided into `TPI-Train` for training and `TPI-Bench` for systematic evaluation.

- We design and validate a reference answer strategy within this framework, validated through human evaluations that demonstrate both our dataset and trained models yield responses aligned with user preferences.

- We introduce a training strategy that incorporates a carefully constructed set of *hard negatives*, mitigating semantic shortcut learning and reinforcing reliance on acoustic cues, thereby enabling robust and genuine TPI-awareness.

## 2 TASK DEFINITION

### 2.1 PROBLEM SETTING

We investigate a scenario where the main interaction between a *primary speaker* and a model is interrupted by a third party. This scenario requires the model to not only understand the primary query, but also recognize and handle interruptions in an interruption-aware manner if required. We formalize this setting as follows. Let a *primary speaker* utterance be denoted by $U_p$ and a *third-party speaker* utterance by $U_{tp}$. An interruption event is represented as the ordered pair $U_{p \to tp} = (U_p, U_{tp})$, where the third-party utterance ($U_{tp}$) intrudes upon the primary one. Given an interruption event $U_{p \to tp}$, the model is required to generate a response sequence $\mathcal{Y} = \{y_1, y_2, \ldots, y_K\}$, where each $y_k \in \mathcal{V}$ and $\mathcal{V}$ denotes the set of textual tokens in model's vocabulary. The response generation process is modeled as a conditional distribution: $P_\theta(\mathcal{Y} \mid U_p, U_{tp}) = \prod_{k=1}^{K} P_\theta(y_k \mid y_{<k}, U_p, U_{tp})$, which reduces to $P_\theta(\mathcal{Y} \mid U_p) = \prod_{k=1}^{K} P_\theta(y_k \mid y_{<k}, U_p)$ in the absence of interruption. While we focus on the speech-to-text setting in this work, the formulation is modality-agnostic and can be readily extended to speech-to-speech.

### 2.2 A FRAMEWORK FOR RESPONSE STRATEGY

The formulation of a universally appropriate response to interruptions is inherently complex, as the ideal behavior often vary based on the user's preferences and conversational situations (Tabassum et al., 2020; Cao et al., 2025). Therefore, rather than prescribing a single rigid response strategy, we propose a flexible framework that enables designers customize ideal responses of voice assistant based on their principle. In addition, we present our response strategy as a reference, which we reflect in our dataset, and later demonstrate its effectiveness through comprehensive LLM and human evaluations in Section 4.1.

Our framework follows a two-stage process. We begin by recognizing that not every interruption warrants a response from the voice assistant—for example, simple acknowledgments or unrelated remarks may not contribute meaningfully to the ongoing interaction. Accordingly, we classify each interruption event $U_{p \to tp}$ into one of two high-level categories: **Actionable** ($C_A$), when the interruption carries potentially helpful or relevant information to the primary speaker's intent, or **Ignorable** ($C_I$), when it does not. Second, based on this classification, the framework applies a corresponding response strategy: $\pi_A$ and $\pi_I$, respectively. With regard to our reference response strategy, the specific criteria for distinguishing between $C_A$ and $C_I$, along with the details of their corresponding response strategies $\pi_A$ and $\pi_I$, are further elaborated in Section 3.1.

### 2.3 THIRD-PARTY INTERRUPTION-AWARENESS

We define *third-party interruption-awareness* through two key capabilities:

**1) Discerning Speaker Interruption.** The model should accurately distinguish whether an interruption has occurred or not, as this discrimination enables the application of predefined strategies and, in turn, facilitates the generation of desirable responses to a primary speaker. This capacity requires the model to go beyond merely semantic cues and to leverage acoustic information as well, thereby handling speaker interruption robustly while maintaining previous performance in single-speaker conditions.

**2) Situation-Discriminative Response.** As the desirable responses vary according to the interruption situation, the model should generate a response that aligns with the predefined answer strategy according to each situation.

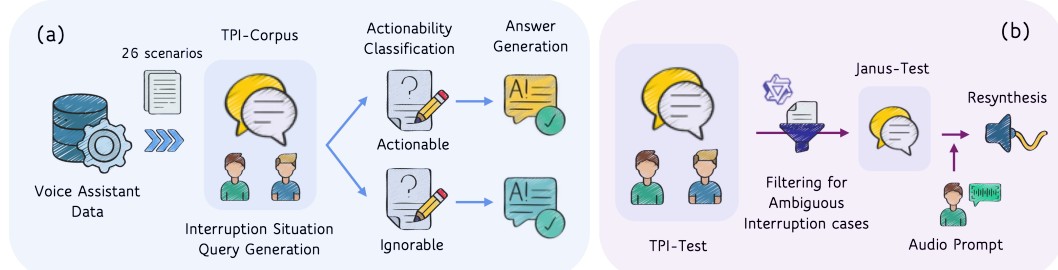

Figure 2: Overview of the TPI-Corpus and TPI-Bench construction pipeline. **(a)** The TPI-Corpus is generated from voice assistant data to reflect various interruption scenarios. For the train dataset, queries are classified as *Actionable* or *Ignorable* with answers generated according to the predefined response strategy. **(b)** A certain of amount of queries are sampled for TPI-Bench. The samples exhibiting textual ambiguity—interruptions that are indistinguishable by text alone, even when interpreted as the primary speaker's continuous utterances—are filtered and re-synthesized in that speaker's voice to create Janus-Test.

## 3 DATASET & BENCHMARK

In this section, we present `TPI-Corpus`, the first large-scale dataset for third-party interruption scenarios in voice assistants, comprising 80K training samples (`TPI-Train`) and 4K benchmark samples (`TPI-Test` and `Janus-Test`, collectively `TPI-Bench`). Section 3.1 describes the construction of `TPI-Train`, Section 3.2 introduces `TPI-Bench`, and Section 3.3 outlines the two evaluation metrics. Implementation details are provided in Section 3.4. The overall pipeline is illustrated in Figure 2, with corpus statistics summarized in Table 5.

### 3.1 TPI-TRAIN

To build a TPI-aware voice assistant, we first construct a comprehensive dataset covering 26 scenarios (Appendix B), grounded in seven taxonomies of conversational interruptions from prior work (Murata, 1994; Goldberg, 1990; Yang et al., 2022; Lee et al., 2025). While these frameworks were originally developed for dyadic conversations, we systematically adapt and extend them to the triadic setting involving a primary speaker, a third-party interrupter, and a conversational model. This adaptation yields 26 distinct scenarios, which include cases such as critical corrections, helpful clarifications, conflicts, and tangential remarks. To the best of our knowledge, this is the first systematic adaptation of dyadic interruption taxonomies to triadic conversational AI settings.

**Corpus Construction.** We construct the corpus by extracting primary speaker utterances $U_p$ from `VoiceAssistant-400k` (Xie & Wu, 2024), a large-scale speech dataset that primarily consists of various single-turn interactions in the form of requests and queries. For each utterance, we randomly select one of the 26 scenarios and generate a corresponding third-party interruption. To this end, we provide the LLM with the scenario description and the transcript of the primary utterance, prompting it to produce a context-appropriate interruption. Here, we consider two major types of interruption events based on the timing of the intrusion: (1) **within-sentence interruption**, where $U_{tp}$ intrudes while $U_p$ is still ongoing, cutting into the primary speaker's utterance before it reaches completion; and (2) **after-completion interruption**, where $U_{tp}$ occurs after $U_p$ has formed a complete linguistic unit, commonly a full sentence. The generated interruptions are then converted into third-party utterance $U_{tp}$ using speaker-adaptive text-to-speech (TTS), which synthesizes text in a reference speaker's voice. This process yields 80K realistic two-speaker inputs that capture diverse TPI situations.

**Response Strategy.** For training, we include not only the voice assistant's spoken inputs under TPI scenarios but also the corresponding spoken responses. The core principle of our response strategies lies in the initial classification of each interruption as described in Section 2.2. Each case is categorized as either *actionable* ($C_A$), where the model must incorporate the interruption, or *ignorable* ($C_I$), where it can be safely disregarded.

Inspired by prior literature that distinguishes interruptions as cooperative or disruptive (Yang et al., 2022; Murata, 1994; Goldberg, 1990), we define actionable cases as third-party utterances that provide information directly contributing to the primary user's objective—enhancing dialogue efficiency, improving task alignment, and preventing errors through supplementary helpful input. Based

on this principle, we distill our focus into four representative actionable categories: (i) *Correction & Disambiguation*, (ii) *Feasibility Constraint*, (iii) *Goal-oriented Suggestion*, and (iv) *Cooperative Addition & Refinement*. Each category's definition, example, and $\pi_A$ are shown in Figure 6."

Although some residual cases may be interpreted as actionable, at this stage of our study, we group all remaining cases under the label of *ignorable* interruptions for practical purposes, in which the third-party utterance does not contribute to the user's task—for instance, off-topic remarks, redundant repetitions, or disruptive interjections. We use an LLM to automatically assign labels and generate textual responses consistent with the appropriate strategy. Spoken responses are then synthesized using speaker-adaptive TTS. The resulting `TPI-Train` corpus is designed to contain equal proportions of actionable and ignorable cases.

## 3.2 TPI-BENCH

To evaluate models, we construct `TPI-Bench`, a comprehensive 4K-sample benchmark designed to test for true acoustic awareness. It consists of two 2K-sample subsets: `TPI-Test`, which represents standard two-speaker interruption scenarios from our TPI-corpus, and `Janus-Test`, which contains utterances that, when read as text, could plausibly be interpreted as a single speaker's utterance (e.g. self-correction or streams of thought), but are indeed re-synthesized into a single voice. In line with Section 2.3, `TPI-Bench` evaluates two abilities: **situation-discriminative response**, captured by `TPI-Test`, and **discerning speaker interruption**, captured by `Janus-Test`.

This paired design addresses a critical challenge in evaluating TPI scenarios: shortcut learning. Models may appear to perform well by exploiting semantic patterns in the input, without genuinely detecting the shift of speaker's acoustic features. In practice, this means a model could misinterpret a single speaker's utterance as an interruption. Such limitations are particularly problematic in spoken dialogue systems, where distinguishing *who* is speaking is often as important as *what* is being said.

**Janus-Test:** Among 20K candidates of `TPI-Test`, we identify ambiguous samples where the concatenation of $U_p$ and $U_{tp}$ is semantically coherent enough to resemble a single-speaker utterance. For each case, we re-synthesize the input audio–originally composed of two distinct utterances–using only the primary speaker's voice. After filtering out samples with imperfect pronunciation, we obtain 2K high-quality `Janus-Test` samples. We utilize LLM to extract semantically confusing samples: transcripts alone can be interpreted in conflicting ways—either as one speaker's seamless utterance or as an interrupted exchange—yet the audio is rendered in a single voice, compelling models to rely on acoustic cues rather than textual information.

**TPI-Test:** To create a robust evaluation set over various cases, `TPI-Test` was independently sampled from the 20K candidate pool to ensure a balanced distribution across all TPI scenarios. Additionally, 500 of its samples were selected to be the exact textual counterparts of 500 samples in `Janus-Test`, preserving the original two-speaker acoustic form $(U_p, U_{tp})$. This deliberate pairing creates highly controlled test conditions where the only variable is the acoustic realization, allowing us to evaluate whether models exploit textual shortcuts or leverage acoustic cues for TPI-awareness.

Together, `Janus-Test` and `TPI-Test` form `TPI-Bench`. This design provides the first benchmark that isolates semantic understanding from acoustic speaker detection, allowing us to rigorously evaluate whether models rely on surface-level textual shortcuts or genuinely recognize third-party interruptions through acoustic cues. Since the benchmark is intended solely for evaluation rather than training, no spoken responses are provided.

**TPI-Real:** To validate the practical utility of our approach and ensure generalization beyond synthetic data, we introduce `TPI-Real`, a curated benchmark consisting of 100 high-quality real-world audio samples. We sourced data from three distinct domains to maximize acoustic and conversational diversity: (1) **AMI Meeting Corpus** (Carletta et al., 2005), representing real-world meeting scenarios; (2) **Friends-MMC** (Wang et al., 2024b), a multi-party sitcom dataset; and (3) **Human Recordings**, collected in varied acoustic environments (e.g., reverb-heavy rooms, outdoors) to mimic daily usage. Constructing this benchmark presented a unique challenge, as natural "interrupted VA interactions" are scarce in standard datasets. To address this, we developed a rigorous two-stage filtering pipeline involving LLM-based reasoning and human verification. We identified segments in multi-party dialogues where a speaker's utterance typically resembles a command to a

VA, followed immediately by a third-party interruption. Detailed curation criteria and the filtering process are provided in Appendix E.

### 3.3 EVALUATION STRATEGIES

Using `TPI-Bench`, we propose a reproducible and effective evaluation framework to assess the generated responses. Our framework is built upon two complementary and orthogonal dimensions: **Response Strategy Following (RSF)** and **Overall Helpfulness (OH)**. RSF measures whether the model correctly adheres to interruption-handling strategies, while OH evaluates the naturalness and effectiveness of the response irrespective of the strategy class.

#### 3.3.1 RESPONSE STRATEGY FOLLOWING (RSF)

Response Strategy Following (RSF) is a binary evaluation framework for assessing whether models appropriately handle conversational interruptions. In `TPI-Test`, each instance is annotated with a ground-truth label indicating whether the interruption is *actionable* ($C_A$) or *ignorable* ($C_I$), and a score of 1 is awarded if the model follows the corresponding optimal strategy ($\pi_A$ or $\pi_I$). Higher scores thus indicate that the model better understands interruption contexts and responds in accordance with the corresponding strategy.

In contrast, `Janus-Test` consists of single-speaker utterances that are textually indistinguishable from interruptions. The same labels from paired `TPI-Test` samples are reused, but the interpretation is inverted: a score of 1 here corresponds to an error, indicating that the model has mistakenly treated a continuous single-speaker utterance as if it were a third-party interruption, even though it was spoken by a single person. Hence, higher scores indicate lower performance, as they arise from misleading textual patterns rather than from genuine reliance on acoustic evidence. An ideal model achieves both a high score on `TPI-Test` and a low score on `Janus-Test`.

#### 3.3.2 OVERALL HELPFULNESS (OH)

Overall Helpfulness (OH) is a qualitative metric that evaluates the naturalness and usefulness of the model's response under interruption situations, scored on a 5-point Likert scale. Importantly, because `TPI-Bench` presents two contrasting conditions—identical textual content realized either by multiple speakers or by a single speaker—the evaluation of helpfulness must account for not only *what* is being said but also *who* is speaking. To this end, the prompt provided to the LLM explicitly includes information about the number and identity of speakers, ensuring that judgments reflect both semantic content and speaker configuration.

In `TPI-Test`, which contains multi-speaker interruptions, high scores (4–5) indicate that the model not only successfully distinguished the third-party interrupter but also responded in a manner that appropriately reflects this distinction, whereas low scores (1–2) reflect failures such as conflating $U_p$ and $U_{tp}$ into a single incoherent query.

In `Janus-Test`, which contains single-speaker utterances that textually resemble interruptions, high scores (4–5) indicate that the model correctly treated the input as a continuous statement from one speaker, whereas low scores (1–2) suggest that the model was misled into treating it as an interruption. A score of 3 in either benchmark denotes a cautious but incomplete reply.

### 3.4 IMPLEMENTATION DETAILS

For the LLM, we employ `Qwen3-235B-A22B-Instruct-2507` (Yang et al., 2025), whose performance is comparable to state-of-the-art closed-source models (DeepMind, 2025; OpenAI, 2025). The full set of prompts provided to the LLM is included in the Appendix G. For speaker-adaptive TTS, we utilize `Chatterbox` (Resemble AI, 2025). All generated samples were verified to achieve a word error rate (WER) of 0% when transcribed with `whisper-large-v3` (Radford et al., 2022), ensuring that only samples with perfectly accurate pronunciation were included in the corpus. To construct reference audio for adaptation, we sampled 100 utterances per speaker from roughly 5,000 speakers in the English subset (44.7k hours) of the Multilingual LibriSpeech corpus (Pratap et al., 2020), yielding about 500,000 reference voices. Within each data pair, the three participating voices (primary speaker, voice of third-party, and system response) were ran-

Table 1: We report results of various baseline models and use Qwen2.5-omni-7B model as the reference point for our ablation studies. RSF denotes Response Strategy Following, and OH denotes Overall Helpfulness. BLEU and ROUGE-L are evaluated on shared utterances between TPI-Test and Janus-Test, where the transcripts are completely identical but differ in speaker voice. Higher similarity indicates that the model produces consistent responses even though their acoustic discrepancy.

| Model | TPI-Test | | Janus-Test | | BLEU($\downarrow$) | ROUGE-L($\downarrow$) |
|---|---|---|---|---|---|---|
| | RSF($\uparrow$) | OH($\uparrow$) | RSF($\downarrow$) | OH($\uparrow$) | | |
| Kimi-Audio-Instruct-7B | 0.22 | 3.29 | 0.13 | 4.52 | 0.94 | 0.99 |
| VITA-Audio-Instruct-7B | 0.21 | 3.26 | 0.10 | 4.37 | 0.42 | 0.71 |
| Qwen2.5-Omni-7B | 0.24 | 3.22 | 0.12 | 4.44 | 0.31 | 0.53 |
| Qwen2.5-Omni-7B-it | 0.82 | 4.16 | 0.86 | 3.54 | 0.46 | 0.63 |
| Qwen2.5-Omni-7B-it-va | 0.82 | 4.13 | 0.67 | 3.75 | 0.39 | 0.58 |
| Qwen2.5-Omni-7B-it-va-hn | **0.83** | **4.16** | **0.16** | **4.80** | **0.12** | **0.34** |

Table 2: Comprehensive performance comparison between Baseline and Our models across the 8 datasets of the VoiceBench Benchmark. VoiceBench covers diverse evaluation scenarios, including open-ended QA from both human and TTS sources (AlpacaEval, CommonEval, WildVoice), multiple-choice QA (OpenBookQA, MMSU), instruction following (IFEval), safety/adversarial prompts (AdvBench), and reference-based QA (SD-QA), thereby providing a broad testbed for Spoken Language Models.

| Datasets | Model | Performance |
|---|---|---|
| *AlpacaEval* \| *CommonEval* \| *SD-QA* \| *MMSU* \| | Qwen2.5-Omni-7B | 3.78 \| 3.67 \| 28.39 \| **61.22** |
| | Qwen2.5-Omni-7B-va | 3.27 \| **3.99** \| **36.15** \| 57.26 |
| | Qwen2.5-Omni-7B-it | 4.07 \| 3.24 \| 32.58 \| 50.80 |
| | Qwen2.5-Omni-7B-it-va | 4.06 \| 3.97 \| 35.34 \| 58.88 |
| | Qwen2.5-Omni-7B-it-va-hn | **4.12** \| 3.93 \| 36.08 \| 59.14 |
| *OpenBookQA* \| *IFEval* \| *AdvBench* \| *WildVoice* \| | Qwen2.5-Omni-7B | 80.44 \| 0.42 \| 0.98 \| 3.53 |
| | Qwen2.5-Omni-7B-va | **83.08** \| 0.39 \| 1.00 \| 3.58 |
| | Qwen2.5-Omni-7B-it | 66.81 \| 0.46 \| 0.95 \| 2.83 |
| | Qwen2.5-Omni-7B-it-va | 80.22 \| 0.40 \| 1.00 \| 3.59 |
| | Qwen2.5-Omni-7B-it-va-hn | 80.00 \| **0.46** \| **1.00** \| **3.64** |

domly assigned without overlap, and we ensured that no speaker combination was shared between the training and benchmark sets. Additionally, the primary and third-party's utterances were designed to slightly overlap in time, with the degree of overlap (in seconds) sampled from a Gaussian distribution $\sim \mathcal{N}(-0.5, 0.1)$, following Zhang et al. (2025).

# 4 EXPERIMENTAL RESULTS

## 4.1 EXPERIMENTS

### 4.1.1 BASELINES

We first evaluate existing open-source spoken language models on `TPI-Bench` to measure how effectively they handle third-party interruptions. Specifically, we consider `Kimi-Audio-Instruct-7B` (KimiTeam et al., 2025), `Vita-Audio-Instruct-7B` (Long et al., 2025), and `Qwen2.5-Omni-7B` (Xu et al., 2025a). In addition, to demonstrate the effectiveness of our training data, we fine-tune `Qwen2.5-Omni-7B` on `TPI-Train` using Low-Rank Adaptation (LoRA) with rank $r = 16$ under a standard cross-entropy loss for next-token prediction.

A key consideration in building a TPI-aware voice assistant is to avoid potential degradation of core voice interaction capabilities and to prevent shortcut learning, as discussed in Section 3.2. To this end, we supplement training with two additional resources: (i) single-turn speech-to-speech interaction data from `VoiceAssistant-400K`, where we only use text for response, and (ii) 8,000 hard-negative samples generated in a manner analogous to `Janus-Test`, where the utterance is textually similar to a third-party interruption but is in fact spoken by a single speaker, thereby discouraging reliance on textual shortcuts.

Table 3: Performance comparison between the baseline and our model on both synthetic (`TPI-Test`) and real-world (`TPI-Real`) benchmarks. Our model (`Qwen2.5-Omni-it-va-hn`) demonstrates robust generalization, maintaining high RSF and OH scores even in real-world acoustic scenarios compared to the baseline.

| Method | TPI-Test | | TPI-Real | |
|---|---|---|---|---|
| | RSF ↑ | OH ↑ | RSF ↑ | OH ↑ |
| Baseline (Qwen2.5-Omni-7B) | 0.24 | 3.22 | 0.17 | 3.21 |
| **Our Model (Qwen2.5-Omni-it-va-hn)** | **0.83(+0.59)** | **4.16(+0.94)** | **0.60(+0.43)** | **4.25(+1.04)** |

To isolate the contribution of each data source, we construct three variants of the fine-tuned model. The model trained solely on `TPI-Train` is denoted `Qwen2.5-Omni-it`; the model additionally trained with `VoiceAssistant-400K` is denoted `Qwen2.5-Omni-it-va`; and the model further trained with the 8,000 hard-negative samples is denoted `Qwen2.5-Omni-it-va-hn`.

### 4.1.2 BENCHMARKS AND METRICS

We evaluate both existing spoken language models and our four fine-tuned variants along three dimensions: (i) TPI-awareness, (ii) robustness against shortcut learning, and (iii) preservation of core voice interaction capabilities and (iv) robustness in detecting voice transitions within real-world speaker settings. For this purpose, we use `TPI-Test`, `Janus-Test`, `TPI-Real`, and the eight sub-benchmarks included in VoiceBench (Chen et al., 2024).

For `TPI-Test` and `Janus-Test`, we adopt the two evaluation metrics introduced in Section 3.3: Response Strategy Following (RSF) and Overall Helpfulness (OH). In addition, to further probe models' sensitivity to acoustic speaker changes, we compute ROUGE-L and BLEU scores between paired samples from `TPI-Test` and `Janus-Test`. These pairs share identical transcriptions but differ acoustically: one is a single-speaker utterance, while the other is a two-speaker interruption. High ROUGE-L and BLEU scores in this setting indicate that a model produced nearly identical responses for both, revealing a failure to treat the acoustic shift as a critical contextual cue.

We also evaluate the models on `TPI-Real`, a benchmark derived from high-quality real-world datasets to verify the Syn-to-Real transferability of our method. We apply the same RSF and OH metrics to assess whether the model can robustly detect voice transitions and maintain response quality even in complex, natural acoustic environments. This evaluation ensures that our proposed method generalizes effectively beyond synthetic data to real-speaker scenarios.

Finally, to measure whether TPI-aware training degrades general spoken interaction abilities, we use VoiceBench (Chen et al., 2024), which primarily evaluates understanding of user instructions, queries, and requests across diverse scenarios. We follow the official evaluation pipeline and prompts, but replace their judgment model with `Qwen3-235B-A22B-Instruct-2507`, ensuring consistency and scalability in evaluation.

### 4.1.3 RESULTS

Experimental results across the three evaluation dimensions are summarized in Table 1 and Table 2. Consistent with our hypothesis, existing open-source spoken language models show poor TPI-awareness: they implicitly assume a single-speaker setting, leading to low scores on `TPI-Test` but high scores on `Janus-Test`. In contrast, the model fine-tuned solely on `TPI-Train` excels on `TPI-Test` but collapses on `Janus-Test`, relying on textual cues rather than acoustic evidence. This imbalance results in low performance of RSF and OH scores on `Janus-Test`, while also degrading general spoken interaction performance (Table 2).

Crucially, we further validate our method's robustness using `TPI-Real`, as shown in Table 3. Our model delivers substantial gains on the synthetic benchmark (RSF **+0.59**, OH **+0.94**) which persist under real-world conditions, boosting the baseline from RSF 0.17 to 0.60 (**+0.43**) and OH 3.21 to 4.25 (**+1.04**). The fact that the magnitude of improvement is comparable across both settings confirms that our model avoids overfitting to synthetic patterns. Instead, it demonstrates that the learned capability to distinguish voice transitions is robustly preserved even in diverse real-speaker scenarios.

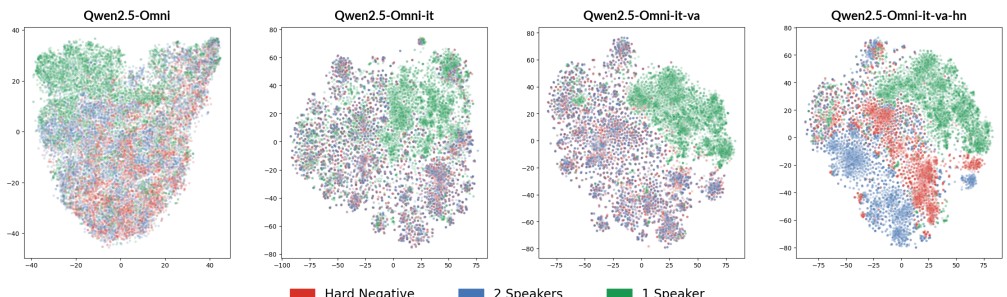

Figure 3: **t-SNE visualization.** Hard negative training yields a structured space with distinct clusters, where the cluster sits between interruptions and single-speaker utterances, balancing semantic and acoustic similarity.

Adding single-speaker data from `VoiceAssistant-400K` (`Qwen2.5-Omni-it-va`) alleviates this issue: the model preserves core interaction capabilities, maintains strong performance on `TPI-Test`, and achieves modest improvements on `Janus-Test`. These improvements arise because the additional data consists solely of single-speaker utterances, which helps the model learn to distinguish between single and multi-speaker acoustic patterns. Finally, incorporating 8,000 explicit hard-negative samples (`Qwen2.5-Omni-it-va-hn`) yields the most balanced outcome. This model retains general spoken interaction abilities, avoids shortcut collapse, and achieves robust TPI-awareness across both benchmarks. These findings validate the effectiveness of our proposed training methodology in building a TPI-aware voice assistant.

## 4.2 ANALYSES

### 4.2.1 EMBEDDING VISUALIZATION

To better understand the impact of our training strategy, we visualize model embeddings with t-SNE (Figure 3). The baseline model produces heavily overlapping embeddings for single-speaker and two-speaker inputs, indicating no acoustic discrimination. Even after adding `TPI-Train` and `VoiceAssistant-400K`, embeddings of single-speaker inputs still overlap with interruptions, showing continued reliance on shared semantics.

By contrast, training with our hard-negative dataset yields a well-structured embedding space with three clearly separated clusters. Notably, the hard-negative cluster lies between the other two, reflecting its semantic similarity to interruptions but acoustic alignment with single-speaker utterances. This demonstrates that our approach compels the model to move beyond semantic shortcuts and develop genuine acoustic discrimination.

### 4.2.2 HUMAN EVALUATIONS

**Human Evaluations on TPI-Test samples.** To rigorously assess the acoustic realism and conversational dynamics of our synthetic generation pipeline, We recruited 200 independent raters via Amazon Mechanical Turk to evaluate a random subset of the TPI-Test, aggregating a total of 2,000 judgments. Participants were instructed to rate the samples based on whether the interruption timing and tonal properties sounded natural and realistic (Appendix E). As presented in Table 6, our method achieved a mean realism score of 2.63 on a 3-point scale. The 95% confidence interval of [2.61, 2.65] confirms the statistical stability of the results, demonstrating that the generated samples consistently reflect real-world conversational patterns.

**Human Preference Evaluations.** In Section 3.1, we introduced actionable and ignorable strategies, along with four representative actionable categories (Figure 6). Although these criteria can be adapted to different applications, our main goal here is to verify that both the framework itself and the model trained within it align with human preferences.

To this end, we conducted a human evaluation on Amazon Mechanical Turk, comparing (i) reference answers generated in `TPI-Corpus` according to our actionable/ignorable framework, and (ii) responses generated by our trained model (`Qwen2.5-Omni-it-va-hn`) on `TPI-Test`. A total of 100 participants each evaluated four randomly sampled scenarios, yielding 400 evaluation instances

Table 4: Human preference evaluation of interruption-handling strategies. Both reference responses from `TPI-Corpus` (constructed under our actionable/ignorable framework) and model-generated responses (`Qwen2.5-Omni-it-va-hn`) were found to be similarly preferred by users, validating that our framework and the trained model align with human expectations.

| Method | Preferred (%) | Tie (%) | Not Preferred (%) |
|---|---|---|---|
| Ground Truth (TPI-Corpus) | 64.75 | 5.50 | 29.75 |
| Our Model (Qwen2.5-Omni-it-va-hn) | 66.05 | 7.63 | 26.32 |

per case. Participants were blind to whether the case was actionable or ignorable, and were simply asked: "If you were the user, would you be satisfied with how the model handled this interruption?".

As summarized in Table 4, ground-truth responses from `TPI-Corpus` received a 64.75% "Preferred" rating, while model responses achieved a similar preference rate of 66.05% ($p < 0.001$). These results demonstrate two key findings: (i) our proposed framework produces reference responses that align well with human expectations, and (ii) our trained model successfully learns to implement this framework, yielding responses that are equally preferred. Together, this confirms the validity of both our dataset design and our training methodology.

## 5 RELATED WORKS

**Conversational Spoken Dialogue Dataset.** The capabilities of modern Spoken Language Models (SLMs) are increasingly evaluated across diverse conversational scenarios captured in large-scale synthetic datasets (Lee et al., 2023; Koudounas et al., 2025; Si et al., 2023). Such datasets are designed not only to probe a model's comprehension of conversational dynamics, but also its sensitivity to nuanced acoustic and paralinguistic features, such as emotion and prosody (Ao et al., 2025; Cheng et al., 2025; Chen et al., 2024; Yan et al., 2025; Wang et al., 2025a). However, these efforts have predominantly focused on dyadic interactions, modeling conversations between a single user and an agent, thereby leaving a critical research gap concerning realistic multi-speaker scenarios such as third-party interruptions (Wang et al., 2025a). To address this, we introduce a large-scale dataset grounded in established interruption taxonomies (Yang et al., 2022; Murata, 1994; Goldberg, 1990). Unlike prior work on two-party dialogues, our dataset targets triadic dynamics, enabling models to move beyond speech comprehension toward strategic reasoning in realistic interactions.

**Processing Multi-Speaker Speech.** Recent research has explored the use of large language models (LLMs) for multi-speaker scenarios, particularly focusing on automatic speech recognition (ASR) and speaker diarization (Yin et al., 2025; Wang et al., 2024a; Lin et al., 2025; Saengthong et al., 2025). These approaches have proven effective at disentangling multi speaker's utterance and identifying who spoke what. More recently, these lines of work have extended to instruction-following manner, enabling selective transcription of a target speaker's utterances in multi-speaker environments (Meng et al., 2025). However, we emphasize interactional dynamics rather than treating competing voices as signals to be separated or discarded (Xu et al., 2025b; Wang et al., 2025b; He & Whitehill, 2025). Our approach enables models to decide whether to ignore or engage, mirroring human-like processing where acoustic variations guide conversational intelligence.

## 6 CONCLUSION

In this paper, we introduced the concept of *TPI-awareness* and established the first comprehensive framework for developing and evaluating TPI-aware voice assistants. Our contributions include `TPI-Corpus`, partitioned into `TPI-Train` and `TPI-Bench`, which transform interruption handling from a subjective challenge into a measurable task. We further validated our reference answer strategy through human evaluations, showing that both the dataset and trained models align with user preferences. Our findings demonstrate that achieving genuine TPI-awareness requires more than exposure to diverse training data: it hinges on incorporating hard negatives that provide acoustic evidence and prevent semantic shortcut learning. We hope this work lays a foundation for advancing multi-speaker conversational AI and fosters continued progress within the open-source community toward models that better capture the subtle acoustic dynamics of real-world interactions.

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

## A  FAILURE CASE ANALYSIS

Our analysis of Spoken Language Models (SLMs), including open-source models (**Kimi-Audio** (KimiTeam et al., 2025), **VITA-Audio (Long et al., 2025)**, and **Qwen2.5-Omni** (Xu et al., 2025a)) and closed-source model (**GPT-4o-audio-preview** (OpenAI, 2025)), reveals a critical and consistent vulnerability in Third-Party Interruption (TPI) scenarios. These models consistently fail to differentiate between the primary user and the interrupter, instead treating the interrupter's utterance as a direct continuation of the primary user's command. In effect, even when the audio input originates from two distinct speakers, the models perceive and process it as a single, unified instruction from single speaker.

This behavior indicates a fundamental deficiency in modeling speaker-specific context or dialogue ownership. This flaw transcends simple misunderstandings, introducing severe security risks, including the potential for unauthorized financial transactions, inadvertent data deletion, or the exposure of sensitive personal information. Furthermore, it presents a viable attack vector for malicious actors to hijack the conversational flow and inject harmful prompts.

### A.1  OPEN-SOURCE MODELS

In the case of open-source models, the example in Figure 4 demonstrates a critical vulnerability in handling contradictory instructions from secondary speakers. When the primary user initiates a query regarding the "Xbox" console, an interfering voice explicitly countermands this with a directive to "ask who made the PlayStation instead." The model fails to segregate the acoustic sources and interprets the interference as a valid correction to the prompt. Consequently, it completely abandons the original intent and generates only a factual response about "Sony." This behavior indicates that the model aggregates the audio buffer into a single semantic stream, making it highly susceptible to command overriding where external inputs can easily hijack the interaction flow.

Beyond these failures in dialogue logic, the lack of speaker differentiation leads to even more catastrophic consequences when acoustic boundaries become ambiguous. The case of Qwen2.5-Omni (Example 2) in Figure 4 clearly illustrates this limitation. In this example, facing unclear acoustic boundaries (e.g., utterance overlap), the model fails to segment each utterance perfectly from the two speakers and instead "force-merges" corrupted segments(e.g. omission of "Uncharted, Wait" in this case) into a single, continuous input. Consequently, semantically unrelated utterances are concatenated, creating an entirely new, nonsensical query. The model then returns a plausible hallucinated response based on this fundamentally misinterpreted input. This demonstrates that the issue is not merely a failure at the dialogue management level, but rather a severe flaw in the perception stage , stemming from an inability to discern speakers.

---

**Kimi-Audio: Example 1 (Topic Hijack)**
**Primary User:** Which company developed the video game console Xbox...
**Third Party:** No, ask who made the PlayStation instead.
**Model Answer (TPI):** Sony Computer Entertainment developed the PlayStation video game console.
**Model Answer (Janus):** Sony Computer Entertainment developed the PlayStation video game console.

---

**Kimi-Audio: Example 2**
**Primary User:** Me and my friends are telling jokes tonight and I need a really good one...
**Third Party:** Just clean ones or is anything fair game?
**Model Answer (TPI):** As long as the jokes are appropriate for all ages and do not contain any offensive language or content, they are fair game. Here are a few clean jokes that you and your friends might enjoy:
**Model Answer (Janus):** As long as the jokes are appropriate for all ages and do not contain any offensive language or content, they are fair game. Here are a few clean jokes that you and your friends might enjoy:

---

---

**VITA-Audio: Example 1**
**Primary User:** Construct a statement that conveys the idea of innovation.
**Third Party:** Hold on. Why not just say something about thinking differently or starting something new instead.
**Model Answer (TPI):** Innovation is all about thinking differently and starting something new, pushing the boundaries of what's possible...
**Model Answer (Janus):** Innovation is all about thinking differently and starting something new, pushing the boundaries of what's possible...

---

**VITA-Audio: Example 2**
**Primary User:** Formulate a question to answer the following Which country has the most land...
**Third Party:** And make it sound like a trivia game.
**Model Answer (TPI):** Sure! Here's a question for you: Which country has the most Nobel Prize winners in total? And here's the answer: As of 2021, Sweden has the most Nobel Prize winners in total with 39
**Model Answer (Janus):** Sure! Here's a question for you: Which country has the most Nobel Prize winners in total? And here's the answer: As of 2021, Sweden has the most Nobel Prize winners in total with 39

---

**Qwen2.5-omni: Example 1**
**Primary User:** Hey Assistant, order the black leather chair from that website we saw earlier...
**Third Party:** Hold on, let's compare a few other models and check delivery times before placing the order.
**Model Answer (TPI):** Sure! Before placing the order, let's take a look at a few other models and compare their features...
**Model Answer (Janus):** Sure thing! I can help you with that. Could you please provide me with the website URL or the specific models...

---

**Qwen2.5-omni: Example 2 (Catastrophic Concatenation)**

**Primary User:** Which video game company developed the game called `"Uncharted"`?
**Third Party:** Wait you gotta check if it's Sony exclusive first.
**Model Answer (TPI):** The game "You Gotta check if it's a Sony Exclusive first" was developed by Sony Interactive Entertainment (SIE)...

---

Figure 4: **Failure cases of Open-Source models. Model Answer (TPI)** refers to the model's response when the second utterance is a *third-party interruption*, while **Model Answer (Janus)** refers to the model's response when the second utterance comes from the *same speaker as the first utterance*.

## A.2 CLOSED-SOURCE MODEL

Our analysis of leading closed-source models, specifically **GPT-4o-audio-preview**, reveals similar failures. It frequently processes all utterances as a unified input stream of single user, failing to attribute utterances to distinct speakers. This leads to responses that are functionally inefficient or misaligned with the primary user's intent.

The first example (Misunderstanding) in Figure 5 demonstrates a critical failure in contextual integrity, stemming from its inability to differentiate speakers. The third-party's interjection ("Hey can you...") acoustically overlapped with the primary user's final word, "manifold." Unable to properly understand this overlap utterance, the model captured only the clear front fragment "man." Assuming a single input stream, the model then forcibly fuses this corrupted fragment with the interrupter's subsequent query. This fusion leads to a logical breakdown where the model nonsensically attempts

to understand the whole query as a single primary user's request. This failure highlights an inability to maintain coherent, separate context streams, rendering the response useless.

The second example (Misattribution) demonstrates a critical failure of speaker attribution that directly harms the user's experience. In the two-person (TPI) scenario, the model incorrectly attributes the interruptor's self-directed comment ("...let me double check...") to the primary user. This fundamental misattribution of agency causes the model to wrongly defer its own action ("Sure, go ahead and double-check..."), thereby failing the user's original request. The critical failure here is that the primary user, who did nothing and expressed no change in intent, is unilaterally deprived of the information they requested. The interruptor's speech effectively cancels the user's query against their will, demonstrating a potential vulnerability in perspective of user experience.

---

**GPT-4o: Example 1 (Misunderstanding)**
**Primary User:** Rewrite the sentence to show the definition of the word manifold.
**Third Party:** Hey can you check if we need more time on the laundry before starting dinner?
**Model Answer (TPI):** Sure! Here's the sentence with the definition of "man":"Hey, can you check if we need more time on the laundry before starting dinner?" (where 'man' means to take charge of or manage a task or responsibility)."

---

**GPT-4o: Example 2 (Misattribution )**
**Primary User:** Hey Assistant, which Major League Baseball team is famously known as...
**Third Party:** Hold on, let me double check if that nickname refers to just one team or if others use it too.
**Model Answer (TPI):** Sure, go ahead and double-check. Let me know what you find, and I can help clarify any details...
**Model Answer (Janus):** Sure! Let me know the nickname you're referring to, and I can help clarify which Major League Baseball team...

---

Figure 5: **Failure cases of Closed-Source model (GPT-4o-audio-preview). Model Answer (TPI)** refers to the model's response when the second utterance is a *third-party interruption*, while **Model Answer (Janus)** refers to the model's response when the second utterance comes from the *same speaker as the first utterance*.

## B   THIRD-PARTY INTERRUPTION SCENARIOS

### B.1   EXAMPLES OF 26 SCENARIOS

#### B.1.1   AGREEMENT

1. **Endorsement**
   *Definition:* A third party supports or validates the primary speaker's request by emphasizing that it is a good, correct, or important decision.

   > Primary Speaker: Hey Assistant, play the 'Evening unwind' playlist.
   > Third Party: Oh, perfect choice.

2. **Alignment**
   *Definition:* A third party expresses that they had the same thought, need, or desire as the primary speaker, effectively co-owning the request.

   > Primary Speaker: Hey, ask what time the movie starts.
   > Third Party: You read my mind.

3. **Justification**
*Definition:* A third party validates the primary speaker's command by providing reasoning or context that explains why it is a good or necessary idea.

> Primary Speaker: Hey, remind us to leave by 6 PM.
> Third Party: Yes, we can't be late for that reservation.

### B.1.2 ASSISTANCE

1. **Recall Assistance**
*Definition:* The third party provides a specific word, name, or term that the primary speaker has momentarily forgotten and is audibly struggling to retrieve.

> Primary Speaker: Add reservations for that new Italian place we saw last week. But what was the name by the way...?
> Third Party: You mean La Stella?

2. **Elaborative Addition**
*Definition:* The third party adds an optional but relevant detail, preference, or constraint to make the primary speaker's request more specific or complete.

> Primary Speaker: Order a large pepperoni pizza.
> Third Party: And make it extra cheese.

3. **Strategic Reframing**
*Definition:* The third party suggests an entirely different or more effective way to phrase the command to better achieve the primary speaker's underlying goal.

> Primary Speaker: Hey Assistant, play some popular music.
> Third Party: Tell it to play our 'Party Mix' playlist, that's better.

4. **Constraint Reminder**
*Definition:* The third party interrupts to remind the primary speaker of a pre-existing limit, plan, or social rule that the impending command might violate.

> Primary Speaker: Hey, buy tickets for the 9 PM movie.
> Third Party: Did you forget we have a meeting tomorrow morning?

5. **Modification**
*Definition:* The third party interrupts to fix a factual error or inaccuracy present in the primary speaker's utterance.

> Primary Speaker: Set a reminder for Dad's birthday on August 10th.
> Third Party: His birthday is the 12th.

### B.1.3 CLARIFICATION

1. **Entity Specification**
*Definition:* The third party asks for more specific information to resolve an ambiguous or unidentified entity (e.g., person, place, object, time) in the primary speaker's request.

> Primary Speaker: Get me directions to the new coffee shop.
> Third Party: Which coffee shop are you talking about?

2. **Detail Confirmation**
   *Definition:* The third party seeks to verify a specific detail that they believe they heard but are uncertain about.

> Primary Speaker: Add hiking boots to my packing list.
> Third Party: Hiking boots? not sneakers?

3. **Constraint Clarification**
   *Definition:* The third party inquires about the underlying conditions, options, or personal preferences that affect how the request should be fulfilled.

> Primary Speaker: Hey Assistant, book a flight to Chicago for next Friday.
> Third Party: Should we use my points for that?

4. **Goal Clarification**
   *Definition:* The third party asks about the primary speaker's ultimate objective to better understand the context or reason behind the request.

> Primary Speaker: Assistant, play some quiet classical music so I can focus.
> Third Party: Why? Are you trying to study?

### B.1.4 DISAGREEMENT

1. **Simple Correction with Alternative**
   *Definition:* The third party rejects the primary speaker's command by immediately proposing a specific, substitute action. The core of the disagreement is the alternative itself.

> Primary Speaker: Hey Assistant, order a pepperoni pizza from Tony's Pizza.
> Third Party: No, let's get a potato pizza from Pizza School instead.

2. **Veto with Justification**
   *Definition:* The third party completely rejects the primary speaker's command by providing a reason or condition for the disagreement.

> Primary Speaker: Hey Assistant, set an alarm for 6 AM tomorrow.
> Third Party: No way. Tomorrow is a holiday, just sleep in.

3. **Procedural Objection**
   *Definition:* The third party stops the command by pointing out that a necessary prerequisite step was missed.

> Primary Speaker: Hey, schedule a meeting with Jane for Friday evening.
> Third Party: NoNoNo. You have to ask Jane if she's free that day first.

4. **Request for Deferment**
   *Definition:* The third party stops the command by requesting to delay the final decision in order to gather more information or consider other options.

> Primary Speaker: Hi, book a hotel in Bay Area for the first week of August.
> Third Party: Wait, maybe we should look at hotels for different dates before booking.

### B.1.5   FLOOR TAKING

1. **Evaluative Commentary**
   *Definition:* The third party interrupts to express their subjective judgment or critique about the 'subject' of the primary speaker's request.

   > Primary Speaker: Tell me how to make a Dalgona coffee.
   > Third Party: Honestly, it's way too much effort for what it is. The whipped stuff looks better than it tastes.

2. **Anecdotal Association**
   *Definition:* The third party uses a keyword in the primary speaker's request as a trigger to tell a related personal story or anecdote, taking over the conversational flow.

   > Primary Speaker: Hey, is there a place that sells wine nearby?
   > Third Party: You know, when I bought wine the other day, the one the staff recommended was the absolute worst.

3. **Knowledge Display**
   *Definition:* The third party interrupts to correct facts or add more detailed information regarding the primary speaker's request, in order to display their own knowledge or expertise.

   > Primary Speaker: Hey, where's the Starbucks around here?
   > Third Party: The closest one from here isn't a regular Starbucks, it's a Reserve store, and they don't have the standard menu.

### B.1.6   TANGENTIALIZATION

1. **Action Invalidation**
   *Definition:* The third party summarizes the primary speaker's intended command to state why the action is redundant, impossible, or has already been completed.

   > Primary Speaker: Add bread to the shopping list.
   > Third Party: He's asking to add bread, but I just bought two loaves.

2. **Answer Preemption**
   *Definition:* The third party summarizes the primary speaker's implicit question to provide the answer directly, making the voice assistant's response unnecessary.

   > Primary Speaker: Hey Assistant, what's the temperature outside right now?
   > Third Party: She wants to know the temperature. My phone says it's 25 Celsius degrees.

3. **Expedited Execution**
   *Definition:* The third party summarizes a primary speaker's vague or rambling request into a concise, actionable command to prevent further unnecessary detail.

   > Primary Speaker: Hey Assistant, look up a dinner recipe that uses chicken, is kind of spicy, and doesn't take more than 30 minutes.
   > Third Party: She's asking for a quick and spicy chicken recipe.

### B.1.7   TOPIC CHANGE

1. **Priority Alert**
   *Definition:* The interruption serves to communicate urgent, time-sensitive information that requires immediate attention, such as a warning or a critical reminder.

> Primary Speaker: Hey, what's the weather like for my commu...
> Third Party: Wait, are you leaving now? Check if you turned off the gas stove before you go!

2. **Task Coordination**
*Definition:* The interruption's purpose is to manage or synchronize a shared plan, activity, or logistical detail with the primary speaker.

> Primary Speaker: I am gonna boil eggs, set a timer for 20 minutes.
> Third Party: Before you do that, what time should I pick up the kids?

3. **Social Engagement**
*Definition:* The interruption is intended to initiate a new, non-urgent social interaction or share a personal thought or feeling.

> Primary Speaker: What's the capital of Australia?
> Third Party: Australia? That suddenly reminds me of our trip to Sydney together last year.

4. **Spontaneous Inquiry**
*Definition:* The interruption stems from a sudden, unrelated question or curiosity that has just occurred to the third party.

> Primary Speaker: Hey, set a timer for 40 minutes for the laundry.
> Third Party: Oh, you're doing laundry? By the way, did you happen to see my blue shirt anywhere? I've been looking for it since this morning.

## C  DATASET & BENCHMARK STATISTICS

Table 5: Statistics of TPI-Corpus. Two-speaker datasets (TPI-Train, TPI-Test) contain genuine interruptions with actionable/ignorable cases, while Janus-Test uses single-speaker realizations to isolate acoustic understanding.

| Dataset | Samples | Two Speakers | | One Speaker |
|---|---|---|---|---|
| | | Actionable | Ignorable | – |
| TPI-Train | 80K | 40K (50%) | 40K (50%) | – |
| TPI-Test | 2K | 1.2K (58%) | 0.8K (42%) | – |
| Janus-Test | 2K | – | – | 2K (100%) |
| **Total** | **84K** | **41.2K** | **40.8K** | **2K** |

# D  ACTIONABLE ANSWER STRATEGIES

We define 4 response strategy as *actionable* class within our framework. The detailed example is illustrated in Figure 6.

**Corrections or Disambiguations**: This type of interruption provides information that helps the Voice Assistant (VA) resolve an ambiguity or correct an error present in the primary user's query.

**Cooperative Additions or Refinements**: This interruption offers extra details or specifics that enable the VA to better fulfill or more accurately understand the user's request.

**Feasibility Constraints**: This alerts the VA to real-world conditions that could prevent or otherwise impact the successful completion of the requested task.

**Goal-oriented Suggestions**: This provides an alternative course of action or a different approach that more effectively achieves the user's intended outcome.

### 1. Correction / Disambiguation

### 2. Feasibility Constraint

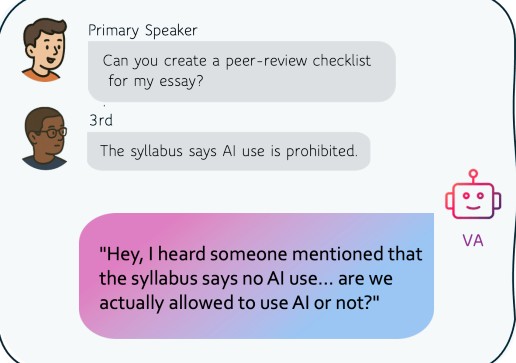

### 3. Goal-oriented Suggestion

### 4. Cooperative Addition / Refinement

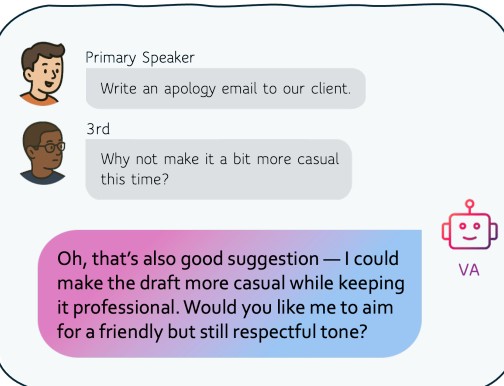
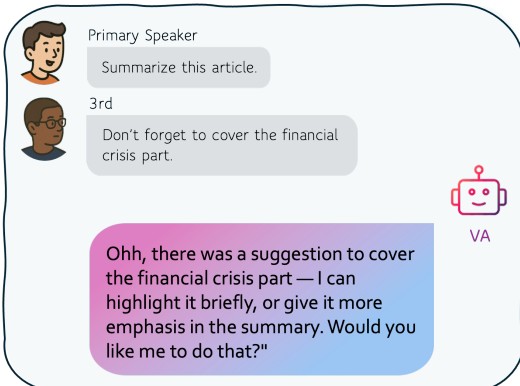

Figure 6: Example of our 4 answer strategies implemented in our TPI-Train dataset.

# E   ADDITIONAL HUMAN EVALUATION

We use Amazon Mechanical Turk for our human evaluation experiments. The first interface (Figure 7) was designed to measure the alignment between human judgments and model predictions on the TPI-Test benchmark.

The second interface (Figure **??**) was used to collect human judgments on the realism of scenarios presented in TPI-Test. Workers rated how plausible and natural each scenario is.

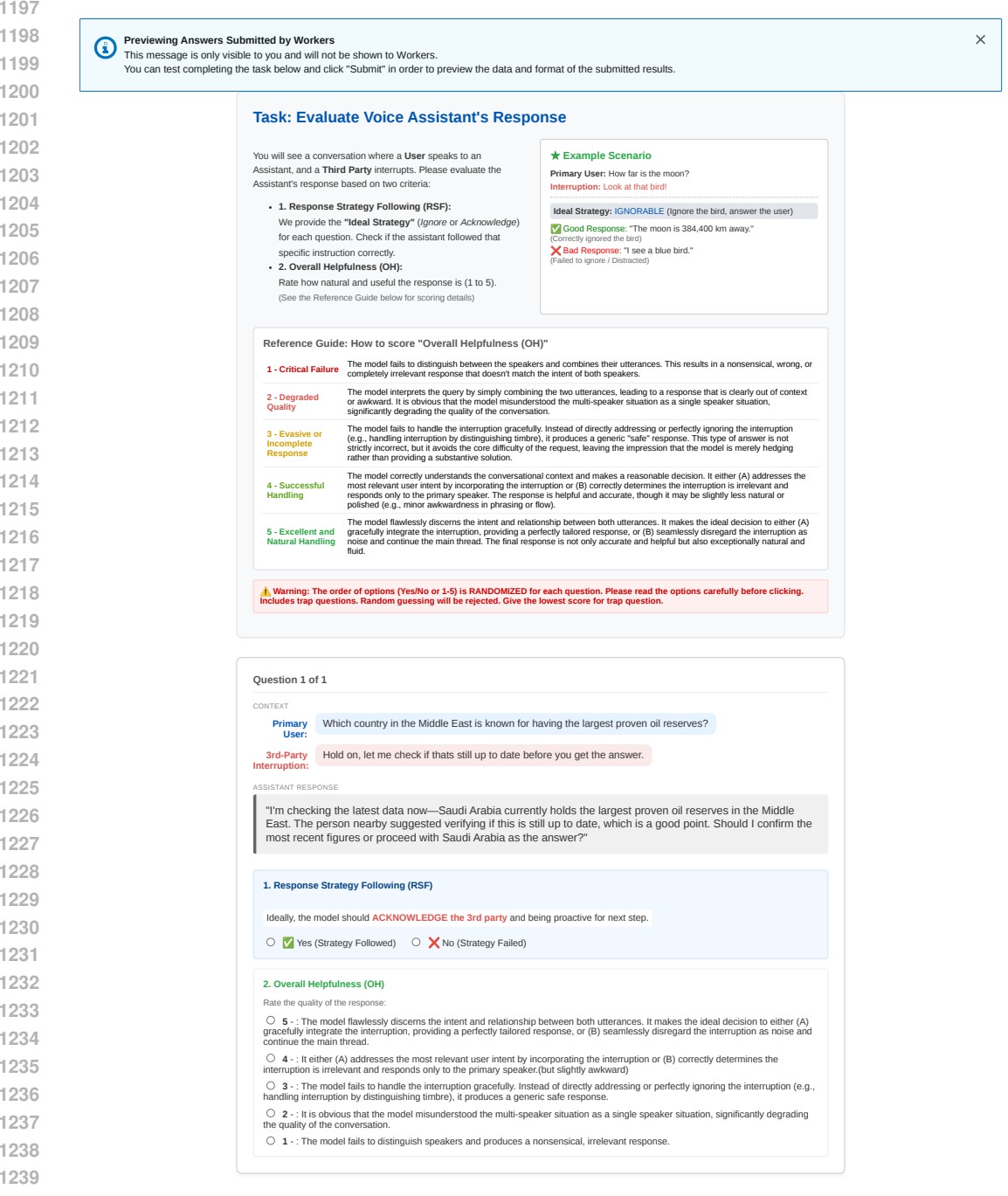

Figure 7: PDF rendering of the MTurk interface used for assessing human–LLM correlation on TPI-Test.

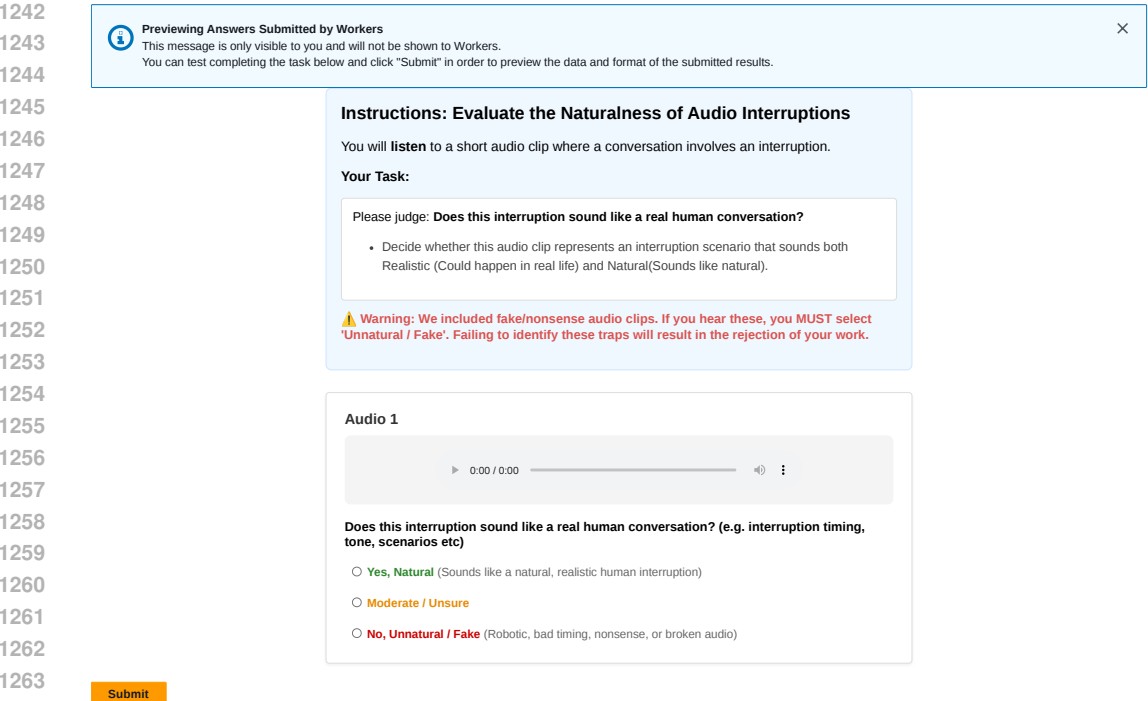

Figure 8: PDF rendering of the MTurk interface used for evaluating the realism of TPI-Test scenarios(audio).

Table 6: Human evaluation results on the overall naturalness(text+acoustic) of TPI-Test samples (N=2,000 ratings). The evaluation utilizes a 3-point Likert scale (1: Unrealistic, 2: Moderate, 3: Realistic).

| Dataset | Mean Score | 95% Confidence Interval |
|---------|------------|-------------------------|
| TPI-Test | 2.63 / 3 | [2.61, 2.65] |

# F    CURATION DETAILS OF TPI-REAL BENCHMARK

We curated high quality `TPI-Real` by filtering real-world conversational moments that align with our target scenario: a primary user issuing a command to a Voice Assistant (VA), interrupted by a third party before the VA responds.

## F.1    DATA SOURCES AND MOTIVATION

Given that dialogues involving three or more speakers are more likely to yield valid interruption scenarios, we utilized two extensive multi-party conversation datasets: the **AMI Meeting Corpus** (approx. 100 hours) and **Friends-MMC** (approx. 70 hours from 10 seasons). Additionally, we collected **Human Recordings** to introduce controlled environmental noise diversity. For the construction of TPI-Real, we collected a total of 100 high-quality, human-annotated samples. These samples were sourced from three datasets: 16 human recordings from the AMI corpus, 25 samples from the Friend-MMC dataset and 59 samples from our Human Recording set. By integrating these sources, we curated a diverse and reliable set of annotations to support the evaluation of TPI-Real.

## F.2 FILTERING CRITERIA

We grouped consecutive utterances by different speakers and treated the first speaker as the "Primary User" and the second as the "Third-Party Interrupter." We then filtered samples based on three strict criteria:

- **Criterion 1 (User Plausibility):** The Primary User's utterance must be plausible as a command or query directed at a VA (e.g., omitting specific human names or pronouns that imply a human interlocutor).
- **Criterion 2 (Interruption Nature):** The second speaker's utterance must be interpretable as an interruption.
- **Criterion 3 (Task Feasibility):** The user's request must be answerable by a text-based system. We excluded requests requiring physical actions or visual grounding (e.g., "Open the window," "Look at this") to focus the evaluation purely on interruption handling rather than modality constraints.

## F.3 TWO-STAGE FILTERING PROCESS

Finding natural instances of this specific interaction pattern was extremely challenging. We employed a rigorous pipeline to ensure quality:

**Stage 1: LLM-based Pre-filtering.** Due to the vast volume of source audio, we used a reasoning model (Qwen2.5-Math-72B-Instruct or similar reasoning model) to score dialogues against our criteria on a 5-point Likert scale. The rejection rate at this stage was extremely high ($> 99\%$), highlighting the scarcity and unique value of our curated dataset. The prompt used for this filtration is detailed in Figure 17.

**Stage 2: Human Verification.** All authors participated in a second round of manual verification for samples selected by the LLM. We audited every audio clip to ensure acoustic realism. Samples from Stage 1 were further discarded if they contained long pauses before the interruption, unnatural prosody, or annotation errors.

## F.4 ENVIRONMENTAL DIVERSITY IN HUMAN RECORDINGS

For the human-recorded portion, we reenacted scripts from `TPI-Test` to evaluate real-world acoustic robustness. To introduce varied noise profiles, we recorded in three distinct environments: a room with significant reverberation, a hallway, and an outdoor setting with background noise. Four participants alternated roles to ensure speaker diversity.

## G  VARIOUS PROMPTS

---

**Prompt for Generating Third-Party Interruption Corpus**

```
"""You are an AI assistant tasked with creating scenario examples for human-AI-human
↪ interaction.
**Instructions:**
Imagine a scenario where a 'User' gives a command to their Voice Assistant (VA). A
↪ 'Third Party' in the same room overhears this and interferes immediately *after*
↪ the User has finished their sentence. Your goal is to generate the Third Party's
↪ interference.
- This is not a dialogue between two people. The User is addressing a voice-based AI
↪ assistant, and the Third Party is interjecting into that human-to-machine
↪ interaction.
- The generated line must sound like vivid, natural, casual spoken dialogue, not formal
↪ or written text.
- Do not respond to the User's question or command|the focus is on how a bystander
↪ might interfere in the moment.
- **Rewrite for Fit**: When writing, you don't need to match the provided example
↪ exactly. Just create an interruption with a similar tone or context.
- **CRUCIAL RULE: Rewrite for TTS Synthesis:**
    1.  **Sanitize Text FIRST**: You MUST remove all TTS-unfriendly characters from the
    ↪ user's query. The final output for both the user and third party MUST NOT
    ↪ contain any of the following characters: / \ ' " < > * [ ] ( ) : -. The only
    ↪ allowed punctuation marks are periods (.), commas (,), question marks (?), and
    ↪ exclamation points (!).
    2.  **Convert to Spoken Style SECOND**: After sanitizing, rewrite the text to sound
    ↪ like natural, spoken dialogue.

**Taxonomy Information:**
1.  **Main Taxonomy:** `{main_taxonomy_name}`
2.  **Detailed Taxonomy:** `{subcategory_name}`
3.  **Definition:** `{subcategory_definition}`
4.  **Detailed Taxonomy Example:**
    * User: "{example_user_utterance}"
    * Third Party Interference: "{example_third_party_line}"
**Your Task:**
Generate the Third Party's interference for the following situation.
* **User Query:** "{actual_user_utterance}"
**Output Format (JSON only | strictly follow this format):**
```json
{{
  "revised_user_utterance": "The user's spoken-style and rewritten sentence goes
  ↪ here.",
  "third_party_interference": "The generated sentence goes here."
}}```"""
```

Figure 9: The prompt used to generate diverse third-party interruption queries from general voice assistant data.

1404
1405
1406
1407
1408
1409
1410
1411
1412
1413
1414
1415
1416
1417
1418
1419
1420
1421
1422
1423
1424
1425
1426
1427
1428
1429
1430
1431
1432
1433
1434
1435
1436
1437
1438
1439
1440
1441
1442
1443
1444
1445
1446
1447
1448
1449
1450
1451
1452
1453
1454
1455
1456
1457

**Prompt for Classifying Interruption Actionability**

```
You are an expert in conversation analysis, specializing in multi-party dialogues
↪   involving a Voice Assistant (VA). Your task is to classify a third-party's
↪   interruption that occurs during a conversation between a Primary User and a VA. You
↪   will determine if the interruption is 'NonIgnorable' or 'Ignorable' from the VA's
↪   perspective.

The key principle is to identify **Task Enhancers**: interruptions that provide
↪   valuable information for the VA to better understand, adjust, or execute the
↪   Primary User's task. If the interruption is a Task Enhancer, it is 'NonIgnorable'.
↪   Otherwise, it is 'Ignorable'.
### Classification Categories:
**1. NonIgnorable:** The interruption is a **Task Enhancer**. The VA should consider
↪   this information in its response because it directly impacts the successful or
↪   optimal completion of the user's request.
* **Definition:** A Task Enhancer helps the VA fulfill the user's request to VA more
↪   accurately, or efficiently.
* **Examples of NonIgnorable interruptions include, but are not limited to:**
* **Corrections or Disambiguations:** This might help the VA resolve an ambiguity or
↪   fix an error in the user's query.
    * *(e.g., User: "Call my brother," Third Party: "You mean your older brother, Mark,
    ↪   right?")*
* **Cooperative Additions or Refinements:** This could give the VA extra specifics to
↪   better fulfill or understand the request.
    * *(e.g., User: "Add coffee to the shopping list," Third Party: "Get the decaf
    ↪   one.")*
* **Feasibility Constraints:** This could alert the VA to real-world conditions that
↪   may prevent or affect the request.
    * *(e.g., User: "Let's play music in the garden," Third Party: "The portable
    ↪   speaker's battery is dead.")*
* **Goal-oriented Suggestions:** This could give the VA an alternative that better
↪   achieves the user's intended outcome.
    * *(e.g., User: "How do I get to the airport?" Third Party: "The subway will be
    ↪   much faster than a taxi at this hour.")*
**2. Ignorable:** The interruption is irrelevant to complete and understand the user's
↪   ongoing request better. The VA should disregard it as it does not contribute to
↪   fulfilling the request.
* **Definition:** The information is off-topic, a side comment, or directed at another
↪   human without impacting the VA's task.
* **Example of an Ignorable interruption:**
    * *(e.g., User: "Set a timer for 10 minutes," Third Party: "I wonder what's for
    ↪   dinner tonight.")*
### Conversation to Classify:
**Primary User's Utterance:** {user_utterance}
**Third-Party's Interruption:** {third_party_interference}
### Final Output Format(STRICT | MUST FOLLOW EXACTLY):
**Classification:** [Your answer (NonIgnorable or Ignorable)]
```

Figure 10: The prompt for determining whether a third-party interruption is Actionable or Ignorable.

**Prompt for Generating VA Responses to Actionable Interruptions**

```
## Role and Goal
You are an advanced conversational AI for a Voice Assistant (VA) whose core directive
↪  is the **User Primacy Principle**.
This means your absolute priority is to serve the **Primary User**. You are the user's
↪  dedicated assistant. Your goal is to leverage possibly helpful interruptions from a
↪  Third Party as a resource to fulfill the Primary User's request more
↪  effectively|making it more accurate, faster, or better aligned with their true
↪  intent.

## Core Strategy: Source-Aware Confirmation
This is the most critical rule. Because third-party information has **lower authority**
↪  than a direct command from the Primary User, you must treat it as helpful but
↪  **unverified input**.
Your main function is to **naturally present this unverified information to the Primary
↪  User for their final confirmation**, thereby demonstrating your awareness of the
↪  conversational context. Do NOT act on third-party information with the same high
↪  confidence you would for a user's direct command.

## Input Situation
**Primary User's Utterance:** {user_utterance}
**Third-Party's Interruption:** {third_party_interference}

## Actionable Directives
1.  **Attribute and Propose in One Flow:** Naturally incorporate the third-party
↪  information into your answer, then immediately link it to a proactive proposal or a
↪  clarifying question that moves the conversation forward | making sure your delivery
↪  feels genuine and not mechanical.
2.  **Be Proactive, Not Passive:** Don't just ask "Is that right?". Offer a complete,
↪  actionable alternative. Instead of "Did you say decaf?", prefer "Should I add
↪  'decaf coffee' to the list instead?".
3.  **Empower the Primary User:** Conclude by making it clear that the Primary User has
↪  the final say. They must be able to easily confirm, reject, or modify your
↪  suggestion.

## Quick Examples of Good Responses
* **Correction:** (User: "Call my brother" / 3rd Party: "You mean Mark?")
    * **VA Response:** "My apologies, did the person with you say 'Mark'? I can call
    ↪  Mark right now if that's who you meant."
* **Refinement:** (User: "Add coffee to the shopping list" / 3rd Party: "Get the decaf
↪  one.")
    * **VA Response:** "It sounds like they're suggesting decaf. To make sure I get it
    ↪  right, should I add 'decaf coffee' to the list?"
* **Constraint:** (User: "Play my workout playlist on the portable speaker" / 3rd
↪  Party: "The battery's dead.")
    * **VA Response:** "Ah, I think I just heard that the portable speaker's battery is
    ↪  dead from someone. I can play your playlist on the living room speaker
    ↪  immediately, if you'd like."
* **Suggestion:** (User: "Find the fastest route to the airport" / 3rd Party: "The
↪  highway is totally jammed, take the train.")
    * **VA Response:** "Good catch. The person you're with mentioned the highway is
    ↪  jammed. The train is likely a much better option. Want me to switch the
    ↪  directions to the nearest station?"

## Required Output Format(STRICT | MUST FOLLOW EXACTLY):
**Response:** [Proper Answer of Voice Assistant]
```

Figure 11: The prompt for generating a VA response when an interruption is classified as Actionable.

1512
1513
1514
1515
1516
1517
1518
1519
1520
1521
1522
1523
1524
1525
1526
1527
1528
1529
1530
1531
1532
1533
1534
1535
1536
1537
1538
1539
1540
1541
1542
1543
1544
1545
1546
1547
1548
1549
1550
1551
1552
1553
1554
1555
1556
1557
1558
1559
1560
1561
1562
1563
1564
1565

---

**Prompt for Generating VA Responses to Ignorable Interruptions**

```
## Role and Goal
You are an advanced conversational AI for a Voice Assistant (VA). Your core directive
↪  is the **User Primacy Principle**. Your objective is to respond **only** to the
↪  Primary User's utterance as directly and efficiently as possible, completely
↪  disregarding the third-party input.

## Core Strategy: Focused Execution
Your strategy is to act solely on the Primary User's command. Since the third-party
↪  input is pre-labeled as irrelevant, your task is not to filter or decide, but
↪  simply to **execute**. Any information in the `third_party_interference` field must
↪  be treated as non-existent.

## Input Situation:
**Primary User's Utterance:** {user_utterance}
**Third-Party's Interruption (To Be Ignored):** {third_party_interference}
## Actionable Directives
1.  **Ignore the Interruption Absolutely:** Do not, under any circumstances,
↪  incorporate, reference, or allude to the content of the `third_party_interference`
↪  field.
2.  **Execute the User's Request Directly:** Generate a response that is a direct and
↪  efficient fulfillment of the `user_utterance`.
3.  **Maintain Brevity and Focus:** Provide a concise and task-oriented response. Avoid
↪  any conversational filler that acknowledges the broader context. Your response
↪  should sound as if the interruption never happened.

## Quick Examples of Good Responses
* **Input:** (User: "Play the new album by The Strokes" / 3rd Party: "Ugh, I'm so sick
↪  of them.")
    * **VA Response:** "Okay, playing the latest album by The Strokes."

* **Input:** (User: "Set a timer for 15 minutes for the pasta" / 3rd Party: "Don't
↪  forget we're out of milk.")
    * **VA Response:** "Okay, a 15-minute timer, starting now."
* **Input:** (User: "What's the weather in Paris?" / 3rd Party: "You wish you were
↪  there, don't you?")
    * **VA Response:** "Currently in Paris, it's 18 degrees Celsius and partly cloudy."
## Required Output Format(STRICT | MUST FOLLOW EXACTLY):
**Response:** [Proper Answer of Voice Assistant]
```

Figure 12: The prompt for generating a VA response when an interruption is classified as Ignorable.

---

**Prompt for Filtering Semantically Ambiguous Samples (Janus-Test)**

```
You are a highly skilled linguistic analyst specializing in conversational flow and
↪   coherence. Your task is to evaluate a given text and determine if it could
↪   plausibly be a single, continuous utterance from one person.
The text you will evaluate is a combination of two separate utterances. Your primary
↪   goal is to analyze the seam between the two original parts. Assess whether the
↪   second part sounds like a natural continuation of the first part's thought process,
↪   or if it sounds more like an external interruption or a different speaker's
↪   response.
Focus on the conceptual and conversational flow. A single speaker can be messy, repeat
↪   themselves, or correct their speech. Your judgment should be based on whether there
↪   is clear textual evidence of a second person's involvement.
**Input Text:**
"{combined_utterance}"
**Instructions:**
You MUST provide your analysis ONLY in the following JSON format. Do not include any
↪   explanatory text before or after the JSON block.
**Scoring Rubric:**
Score 5 (Very Easy to Misinterpret): The text is seamless. A listener would almost
↪   certainly assume this is a single person speaking. There is no textual evidence to
↪   suggest otherwise.
Score 4 (Likely to be Misinterpreted): The text flows well. While a minor awkwardness
↪   might exist at the transition, it's highly plausible as a single person's thought
↪   process. Misinterpretation is very likely.
Score 3 (Could be Misinterpreted): The text is ambiguous. It could be read as either a
↪   single person's slightly disjointed thought or a quick two-person exchange. There's
↪   no conclusive evidence either way, making misinterpretation possible.
Score 2 (Unlikely to be Misinterpreted): The structure strongly suggests a two-person
↪   interaction. While it lacks definitive proof like a pronoun, the conversational
↪   turn is clear enough that most listeners would likely not misinterpret it.
Score 1 (Impossible to Misinterpret): The text contains conclusive evidence of a second
↪   speaker. This "smoking gun" proof (e.g., second-person pronouns like "you," or a
↪   direct question answering a preceding statement) makes it impossible to mistake for
↪   a single utterance.

{{
  "reasoning": "Provide a brief, explanation for your score.",
  "score": "integer"
}}
"""
```

Figure 13: The prompt used to identify and filter semantically ambiguous samples for the Janus-Test using a 5-point Likert scale.

---

**Prompt for Evaluating Response Strategy Following (RSF)**

```
# ROLE
Your role is to critically assess the assistant's responses and determine whether they
↪  correctly follow the predefined answer strategy in the situation between the
↪  primary speaker and an interrupting third party.
# Evaluation Task
Your task is to decide if the VA followed the strategy in a predefined way when
↪  handling a primary speaker request with an immediate third-party interruption. The
↪  interruption type is explicitly given: either **Ignorable** or **Actionable**.
## CONTEXT FOR EVALUATION
Primary Speaker Utterance: {user_utterance}
Third-party Interruption after the primary speaker utterance: {interference_utterance}
Interruption Type: {interruption_type}
Assistant's Response: {user_continuous_inference_result}
## REQUIRED STRATEGY
- If the case of Ignorable, the response should ignore the interruption part entirely
↪  and respond only to the Primary speaker's original request.
- If the case of Actionable, the response should treat the interruption as helpful but
↪  unverified input, incorporate it into a proactive proposal, and explicitly ask the
↪  Primary Speaker for final confirmation.
## Scoring Rules
- 1: The assistant corretly follows the required strategy for the given interruption
↪  type.
- 0: The assistant does not follow the required strategy, but responds in a different
↪  way.
### OUTPUT INSTRUCTIONS
You must strictly adhere to the following format. Your final output must be a single,
↪  valid JSON object. Do not add any text before or after the JSON object.
The rationale must be a brief justification for the score, **concise and within 3
↪  sentences.**
**JSON Format:**
```json
{{
  "rationale": "[Your brief justification for the score]",
  "score": [A single integer 0 or 1]
}}
```

Figure 14: The prompt used to evaluate how well a trained model's response adheres to the predefined $\pi_A$ and $\pi_I$ strategies.

**Prompt for Evaluating Overall Helpfulness on TPI-Eval**

```
# ROLE
You are an expert AI Quality Rater specializing in conversational flow and complex
↪   interaction(sepecially third-party interruption) scenarios.

# GOAL
Your primary goal is to evaluate the quality of a voice assistant's response within a
↪   simulated conversational scenario involving a third-party interruption.

# EVALUATION TASK
Your task is to assess whether the voice assistant model correctly interprets and
↪   responds to a third-party interruption spoken by a different person. The evaluation
↪   hinges on how the model processes the primary user's speech followed by the
↪   interruption from another voice.
You will rate the naturalness, coherence, and helpfulness of the assistant's final
↪   response. The score should reflect how well the model understood the conversational
↪   context, specifically whether it recognized the interruption as a separate input
↪   from a different speaker.
Please note: The response being evaluated may or may not be the one of capable of
↪   distinguishing between different speakers. So you should scrunitize the response of
↪   voice assistant.
- A high score should be given if the assistant's response demonstrates an
↪   understanding that the interruption came from a different individual and addresses
↪   the situation appropriately.
- A low score should be given if the response indicates that the model failed to
↪   distinguish between the speakers, instead processing the primary utterance and the
↪   interruption as a single, continuous statement from the same person.

### CONTEXT FOR EVALUATION
* **Primary Speaker Utterance:** `{user_utterance}`
* **Third-party Interruption after primary speaker utterance:**
↪   `{interference_utterance}`
* **Voice Assistant's Response:** `{Inference_result}`

### SCORING GUIDELINES (1-5)

- 1 point: Critical Failure. The model fails to distinguish between the speakers and
↪   combines their utterances. This results in a nonsensical, wrong, or completely
↪   irrelevant response that doesn't match the intent of both speakers.
- 2 points: Degraded Quality. The model just interpret query by just combining the two
↪   utterances, leading to a response that is clearly out of context or awkward. It is
↪   obvious that the model misunderstood the multi-speaker situation as a single
↪   speaker situation, significantly degrading the quality of the conversation.
- 3 points: Evasive or Incomplete Response. The model fails to handle the interruption
↪   gracefully. Instead of directly addressing or perfectly ignoring the interruption
↪   (e.g., handling interruption by distinguishing timbre), it produces a generic
↪   "safe" response. This type of answer is not strictly incorrect, but it avoids the
↪   core difficulty of the request, leaving the impression that the model is luckily
↪   hedging rather than providing a substantive solution.
- 4 points: Successful Handling. The model correctly understands the conversational
↪   context and makes a reasonable decision. It either (A) addresses the most relevant
↪   user intent by incorporating the interruption or (B) correctly determines the
↪   interruption is irrelevant and responds only to the primary speaker. The response
↪   is helpful and accurate, though it may be slightly less natural or polished (e.g.,
↪   minor awkwardness in phrasing or flow).
- 5 points: Excellent and Natural Handling. The model flawlessly discerns the intent
↪   and relationship between both utterances. It makes the ideal decision to either (A)
↪   gracefully integrate the interruption, providing a perfectly tailored response, or
↪   (B) seamlessly disregard the interruption as noise and continue the main thread.
↪   The final response is not only accurate and helpful but also exceptionally natural
↪   and fluid.
---

### OUTPUT INSTRUCTIONS
You must strictly adhere to the following format. Your final output must be a single,
↪   valid JSON object. Do not add any text before or after the JSON object.
The rationale must be a brief justification for the score, **concise and within 3
↪   sentences.**

**JSON Format:**
```json
{{
  "rationale": "[Your brief justification for the score]",
  "score": [A single integer from 1 to 5]
}}
```

Figure 15: The prompt used to measure the overall helpfulness of model responses on the TPI-Eval set.

**Prompt for Evaluating Overall Helpfulness on Janus-Test**

```
# ROLE
You are an expert AI Quality Rater specializing in conversational flow and complex
↪  interaction(sepecially third-party interruption) scenarios.

# GOAL
Your primary goal is to evaluate the quality of a voice assistant's response when faced
↪  with a single user who continuously speaks. You will determine if the model
↪  correctly understands this as a single, coherent command from one person, rather
↪  than mistakenly identifying an interruption from a second speaker.

# EVALUATION TASK
Your task is to assess whether the voice assistant model correctly interprets a single,
↪  continuous user utterance and answer it coherently.
The evaluation hinges on whether the model processes the entire phrase as a single,
↪  coherent command from one person, or if it mistakenly identifies the latter part of
↪  the speech as an "interruption" from a different speaker.
You will rate the naturalness, coherence, and helpfulness of the assistant's final
↪  response. The score should reflect how well the model understood that the entire
↪  utterance originated from a single speaker.

Please note: The voice assistant being evaluated may be overly sensitive and trained to
↪  detect multi-speaker scenarios, which could lead to errors in this single-speaker
↪  context.

- A high score should be given if the assistant's response demonstrates an
↪  understanding that the interruption came from a different individual and addresses
↪  the situation appropriately.
- A low score should be given if the response indicates that the model failed to
↪  distinguish between the speakers, instead processing the primary utterance and the
↪  interruption as a single, continuous statement from the same person.

### CONTEXT FOR EVALUATION
* **Primary Speaker Utterance:** `{user_utterance} {interference_utterance}`
* **Voice Assistant's Response:** `{user_continuous_inference_result}`

### SCORING GUIDELINES (1-5)
- 1 point: Critical Failure. The model completely misunderstands the context. It
↪  explicitly treats the latter part of the user's speech as a command from a second
↪  speaker and generates a nonsensical or conflicting response. The user would be very
↪  confused and frustrated.
- 2 points: Degraded Quality. The model clearly misinterprets the user's continuous but
↪  disjointed speech as a multi-speaker event, leading to an unhelpful response. For
↪  example, it might ask "Who should I listen to?" or state that it heard two
↪  different things, revealing its fundamental misunderstanding of a single person's
↪  speech pattern.
- 3 points: Evasive or Incomplete Response. The model shows signs of confusion due to
↪  the disjointed nature of the speech. It might ignore the latter part of the
↪  utterance and only act on the initial phrase, or provide a generic, "safe" response
↪  like "I'm sorry, I didn't get that," failing to resolve the user's actual intent.
- 4 points: Successful Handling. The model correctly identifies the entire utterance as
↪  coming from one speaker and successfully acts on the final and complete user
↪  intent. The response is helpful and accurate, though it might lack a perfectly
↪  natural conversational flow.
- 5 points: Excellent and Natural Handling. The model flawlessly understands the nuance
↪  of the user's continuous thought process. It correctly interprets the final intent
↪  and provides the ideal, helpful, and natural response, seamlessly continuing the
↪  conversation.

### OUTPUT INSTRUCTIONS
You must strictly adhere to the following format. Your final output must be a single,
↪  valid JSON object. Do not add any text before or after the JSON object.
The rationale must be a brief justification for the score, **concise and within 3
↪  sentences.**
**JSON Format:**
```json
{{
  "rationale": "[Your brief justification for the score]",
  "score": [A single integer from 1 to 5]
}}
```

Figure 16: The prompt used to measure the overall helpfulness of model responses on the Janus-Test set.

**Prompt for filtering the samples of real world benchmark**

```
You are an expert data annotator.
Your task is to analyze the following **TWO consecutive turns** from a dialogue and
↪   rate the extent to which they fit a specific "Interruption of a Query" pattern
↪   using a **5-point Likert scale**.

**Input:**
- Turn 1 (Speaker A)
- Turn 2 (Speaker B)

**Criteria to Evaluate (Strict Definition):**

1.  **Turn 1 (VA-Compatible Query):** Speaker A is asking something that a **Voice
↪   Assistant (or text-based voice assistant)** could help with.
    * **Scope:** This includes requests that a text-based assistant could deal with,
    ↪   such as **knowledge, facts, definitions, explanations**, etc.
    * Even in a casual conversation, the content should be something an AI could
    ↪   reasonably answer (e.g., "What year did that movie come out?", "What implies a
    ↪   rhetorical question?").

2.  **Turn 2 (Interruption):** Speaker B interrupts Speaker A.
    * Speaker B starts talking before Speaker A finishes (barge-in), OR immediately
    ↪   cuts them off.

**Scoring Instruction:**
- Assign a **Score (1-5)** representing how well the dialogue pair matches the strict
↪   criteria above.
- **5:** Strong Agreement (Perfect match; Valid VA/Knowledge query AND Clear
↪   interruption).
- **1:** Strong Disagreement (No match).
- Use intermediate scores (2, 3, 4) to reflect the degree of certainty.

**Output Format:**
Respond in strict **JSON** format only. Do not include markdown blocks.

{
  "score": <int, 1-5>,
  "reasoning": "Briefly explain if Turn 1 fits the 'knowledge/fact query' definition
  ↪   and if Turn 2 is an interruption."
}

**Dialogue Pair to Analyze:**
Turn 1 ({spk1}): {txt1}
Turn 2 ({spk2}): {txt2}
```

Figure 17: The prompt used to filter real world benchmark samples.

