# OpenReview forum: "TPI-VA: Third-Party Interruption-Aware Voice Assistant"
_ICLR.cc/2026/Conference — ICLR 2026 Conference Withdrawn Submission_

### Official Review · Reviewer_ThYP · 2025-10-26

**Soundness:** 3
**Presentation:** 3
**Contribution:** 3
**Rating:** 4
**Confidence:** 4

**Summary:**

The study's major success lies in establishing the first comprehensive TPI-Awareness framework and developing the innovative Janus-Test benchmark, which rigorously isolates acoustic perception to prevent semantic shortcut learning. The use of a composite training strategy with Hard-Negatives is validated as effective in achieving genuine acoustic discrimination, with the final model's responses being highly aligned with user preferences. However, the study's weaknesses center on generalization and depth: the current response strategy diversity is limited to "actionable" and "ignorable" categories; the reliance on synthesized speech for the TPI-Corpus may restrict the model's robustness against complex, real-world acoustic dynamics; and the system lacks demonstrated capability in handling long-term dialogue context.

**Strengths:**

1. **Novel and Comprehensive Framework for TPI-Awareness:** The paper introduces and formalizes the concept of **TPI-awareness** (Third-Party Interruption-Awareness), establishing the first holistic framework for tackling this crucial, yet underexplored, real-world challenge in conversational AI. The definition, based on two core capabilities (*Discerning Speaker Interruption* and *Situation-Discriminative Response*), provides a clear and actionable path for future research.

2. **Creation of Rigorous Benchmarks to Isolate Acoustic and Semantic Cues:** The authors develop **TPI-Bench** (comprising TPI-Test and the innovative **Janus-Test**) specifically to diagnose the critical failure mode of "shortcut learning." The Janus-Test is a strong methodological contribution as it cleverly forces models to rely on **acoustic evidence** rather than misleading textual cues, enabling a truly robust evaluation of TPI-aware models.

3. **Effective Training Strategy Validated by Ablation and Mechanism Analysis:** The work proposes a novel **composite training approach** that incorporates carefully constructed **Hard-Negatives**. Experimental results, particularly the successful performance on the Janus-Test and the visualization of clearly separated embedding clusters, provide compelling evidence that this strategy is effective in mitigating semantic shortcut learning and enforcing genuine acoustic discrimination.

**Weaknesses:**

1. **Limited Strategy Diversity, Lack of Richer Response Modes:** Although the paper proposes the binary framework of "Actionable" ($C_A$) and "Ignorable" ($C_I$), this categorization of response strategies is still relatively limited. In complex real-world multi-party dialogues, VAs may require richer response modes such as **seeking confirmation from the primary speaker**, **temporarily maintaining silence**, **logging the third-party input without immediate action**, or **explicitly asking for clarification of intent**. The current framework lacks exploration and benchmarking for these more nuanced, socially sophisticated response modes, limiting the model's practical utility and naturalness in complex human-AI interactions.

2. **TPI-Corpus Reliance on Synthesized Speech, Limited Acoustic Realism:** The entire TPI-Corpus and TPI-Bench are built using **speaker-adaptive Text-to-Speech (TTS)**. While WER verification ensures high transcription quality, synthesized speech often fails to capture the full acoustic complexity of genuine human-human interruptions (e.g., shifts in emotion, accelerated speaking rate during overlaps, spontaneous dynamics). This limitation may cause the model's performance to degrade when generalizing to the subtle acoustic cues present in **real-world, unscripted speech**.

3. **Lack of Capability in Handling Long-Term Dialogue Context:** The paper primarily focuses on the immediate handling of **single interruption events** ($U_{p \to tp} = (U_p, U_{tp})$). However, in practical applications, a third-party interruption might be a persistent issue or require reference to **earlier dialogue history**. The model does not demonstrate the ability to **maintain third-party identity and context** across multiple turns (e.g., remembering a constraint set by the third-party in an earlier turn). A benchmark for evaluating TPI-awareness in complex, long-term multi-party dialogues is needed.

**Questions:**

Please refer to weaknesses.

---

> ### Author Response · Authors · 2025-11-24
> **Author Response (1/n)**
>
> Thank you for your valuable and constructive feedback. Please find my detailed responses to each of your comments below.
> ***
> **[W1] Limited Strategy Diversity, Lack of Richer Response Modes**
>
> We appreciate the reviewer’s insightful comment and fully agree that, in real-world settings, a rich spectrum of interaction strategies may exist beyond the two categories discussed in the paper. Indeed, highly capable assistants might choose to acknowledge an interruption, briefly pause, or ask for clarification rather than fully ignoring it. We recognize that such nuanced conversational behaviors can vary across individuals, cultures, personalities, and usage contexts.
>
> In our work, however, the goal is to propose a general and flexible framework rather than to prescribe a single “correct” conversational policy. As defined in Line 144, we classify an interruption as ignorable only when the assistant completely disregards the third-party speech. All other behaviors(a complementary set)—including acknowledgments, confirmations, partial pauses, or brief handling of the interruption—fall into the broader category of actionable in our framework . In other words, our binary categorization is not intended to constrain the space of strategies; rather, it provides a minimal conceptual foundation upon which many nuanced actionable behaviors can be organized. As we illustrated example strategies in Appendix D, the other strategies similar to your suggestion(e.g. seeking confirmation, temporarily maintaining silence) could be customized as “Actionable” by end-users within our frameworks.
>
> We believe that determining which actionable strategy is “optimal” in a given situation is inherently subjective and deeply dependent on human preferences. Instead of enforcing a single normative reference, our objective is to offer a versatile framework that end-users can adapt and personalize according to their conversational expectations. The reference strategies we demonstrate empirically serve as an example—evidence that the proposed framework can support effective model behavior, not a restriction on the broader space of interaction dynamics.
>
> Thus, we respectfully argue that the binary formulation does not limit exploration but instead enables users and future work to build richer, personalized, and context-sensitive interruption-handling strategies on top of a simple, coherent foundation.
>
> ***

---

> ### Author Response · Authors · 2025-11-24
> **Author Response (2/n)**
>
> **[W2] TPI-Corpus Reliance on Synthesized Speech, Limited Acoustic Realism.**
>
> To further validate the practical utility of our TPI-Corpus and ensure our model generalizes well to real-world scenarios, we constructed a new "Real-world TPI Benchmark" consisting of 100 high-quality audio samples. We sourced 3 types of dataset for diversity in acoustic environments and conversational contexts. **1. AMI Meeting Corpus [2]:** Represents real-world meeting scenarios with varying microphone distances (approx. 100 hours). **2. Friends-MMC [1]:** A sitcom dataset (10 seasons, 220 episodes) characterized by multi-party dialogues and frequent interruptions. **3. Human Recordings:** Manual human recordings to mimic daily Voice Assistant usage. The results and analyses are described below and real interaction audio samples are uploaded on the demo page.
>
> **Analysis.**
> Table 1 highlights that the improvements achieved by our model extend consistently from synthetic to real-world settings. On the synthetic benchmark, Qwen2.5-Omni-7B-it-va-hn(our model) delivers substantial gains over the baseline, improving RSF from 0.24 → 0.83 (+0.59) and OH from 3.22 → 4.16 (+0.94). Crucially, these gains persist even under real-world acoustic conditions: compared to the baseline real-world performance (RSF 0.17, OH 3.21), our model achieves RSF 0.60 (+0.43) and OH 4.25 (+1.04).
> These results show two key findings: (1) real-world performance is consistently higher than the baseline by a large margin, and (2) the magnitude of improvement is comparable to the gains observed in the synthetic setting. This directly addresses concerns that the model may overfit to synthetic patterns—the improvements are not only preserved but remain robust across diverse real-world environments, speakers, and noise conditions.
>
> Overall, the strong baseline-relative gains in both synthetic and real-world evaluations validate that models trained on the TPI-Corpus generalize effectively and provide practical utility beyond controlled synthetic scenarios.
>
> **Table 1. Results on high quality real world TPI benchmark**
> | Model                       | TPI-RSF↑ | TPI-OH↑  |
> |-----------------------------|------|------|
> | Qwen2.5-Omni-7B (syn)    | 0.24 | 3.22 |
> | Qwen2.5-Omni-7B-it-va-hn(syn)    | 0.83 | 4.16 |
> | Qwen2.5-Omni-7B (real)    | 0.17 | 3.21 |
> | Qwen2.5-Omni-7B-it-va-hn(real)    | 0.60 | 4.25 |
>
> **[Detailed Benchmark Curation Process]**
>
> **Curation Process:** We employed a two-stage filtering process to identify "Voice Assistant (VA) interaction scenarios interrupted by a third party." First, we used a Qwen-235B-Reasoning model (Qwen-235B) to filter dialogues where a user speaks to a VA followed immediately by an interruption with 5 scale likert rubric score. Second, the authors manually verified all filtered scripts to ensure logical consistency and realistic occurrence in terms of voice assistant scenarios.
>
>     Observation: Despite the vast size of AMI and Friends-MMC, the filtering ratio(LLM, human both) was extremely high and finding natural instances of "interrupted VA interactions" in any multi-party conversation was extremely challenging. We believe this scarcity demonstrates the significant value and necessity of our synthetic TPI-Corpus for training robust models.
> **Environmental Diversity (Human Recordings):** For the human-recorded portion, we reenacted scripts from TPI-Test to test real world acoustic robustness. We selected three distinct environments to introduce varied noise profiles: a room with reverberation, a hallway, and an outdoor setting with background noise. Also we gathered 4 people to record. And they change the role for each script for diversity. The audio samples used for this benchmark and the corresponding inference results are available on our project demo page.

---

> ### Author Response · Authors · 2025-11-24
> **Author Response (3/3)**
>
> **[W3] Lack of Capability in Handling Long-Term Dialogue Context:**
>
> We appreciate the reviewer’s insightful comment regarding long-term dialogue context. We fully agree that for real-world applications, handling persistent third-party constraints (e.g., remembering a restriction set in an earlier turn) is the ultimate goal. However, we posit that effective long-term constraint tracking is predicated on the robust handling of atomic interruption events. If a model fails to distinguish who is speaking at any given moment due to reliance on semantic shortcuts, it inevitably leads to error propagation in tracking speaker identity over time.
>
> Crucially, extending TPI to multi-turn scenarios is fundamentally distinct from merely expanding context in dyadic dialogues. It introduces a multifaceted combinatorial space involving such as **dynamic role switching** (e.g., a primary user shifting to an interrupter role), **varying interruption frequencies**, and **the much more complex definition of actionable strategies for persistent interruptions** and **interruption handling entangled with long-term content**—also factors that inevitably lead to a substantial expansion of evaluation dimensions. Given that even multi-turn modeling in dyadic settings is often treated as a standalone research domain due to its complexity [1][2], we believe that rigorously addressing these triadic dynamics in multi-turn settings merits a dedicated investigation of its own. Considering our comprehensive contributions—formalizing the TPI task, identifying the critical "semantic shortcut" limitation with the solution of composite training approach, and providing the TPI-Corpus/Benchmarks with evaluation strategies along with a customizable framework—we believe our work establishes the essential foundation required to tackle these complex future challenges. Our focus on "Acoustic” TPI-Awareness provides the necessary "integrity" that any future long-term system must rely upon.
>
> Taking a step further, motivated by your insight, to verify whether our model is stable enough to serve as this foundation, we conducted additional experiments to test its robustness within a multi-turn context.
>
>
> **[Experiment Setup]**
>
> We randomly sampled 1,000 instances from the TPI-Test and Janus-Test evaluation sets. To simulate a realistic multi-turn context, we utilized an LLM to generate natural preceding dialogue history, extending the conversation up to maximum 5 turns. To ensure acoustic consistency, we synthesized the preceding user turns using the exact same voice reference and TTS model. We then evaluated whether the model could successfully detect interruptions (or lack thereof) based on vocal cues within this extended context.
>
> **[Analysis]**
>
> **TPI-Test:** In the multi-turn setting, as described in **Table 2**, our model (Qwen2.5-Omni-7B-it-va-hn) demonstrates improved performance across metrics compared to the baseline, effectively identifying interruptions.
>
> **Janus-Test:** The model demonstrates a low Janus-RSF score, correctly recognizing single-speaker scenarios without hallucinating a third party, even with added dialogue history.
>
> Comparing these results with the single-turn performance in **Table 3** (extracted from our main paper), we observe a consistent trend. The model's ability to make accurate decisions mirrors its single-turn capability, confirming that it relies on robust vocal cues rather than being confused by the extended text context. These findings suggest that our proposed "Composite Training Approach" is highly effective not only in single-turn but also in multi-turn environments. We believe this validates our framework as a reliable reference and a solid foundation for future research expanding into more diverse and complex multi-turn scenarios.
>
>
>
> **Table 2. Performance improvement with multi-turn context**
> | Model | TPI-RSF↑ | TPI-OH↑ | Janus-RSF↓ | Janus-OH↑ |
> |-----------------------------|------|------|------|------|
> | Qwen2.5-Omni-7B(baseline) | 0.16 | 3.23 | 0.12| 4.65 |
> | Qwen2.5-Omni-7B-it-va-hn(ours) | 0.81 | 4.37 | 0.16 | 4.90 |
>
> ***
>
> **Table 3. Performance improvement with single-turn context**
> | Model | TPI-RSF↑ | TPI-OH↑ | Janus-RSF↓ | Janus-OH↑ |
> |-----------------------------|------|------|------|------|
> | Qwen2.5-Omni-7B(baseline) | 0.24 | 3.22 | 0.12| 4.44 |
> | Qwen2.5-Omni-7B-it-va-hn(ours) | 0.83 | 4.16 | 0.16 | 4.80 |
>
>
> ***
> Once again, we really appreciate giving us the opportunity to further strengthen our paper.
>  ***
>
>
> **References**
> ***
> [1] Heeseung Kim, Che Hyun Lee, Sangkwon Park, Jiheum Yeom, Nohil Park, Sangwon Yu, & Sungroh Yoon. (2025). Does Your Voice Assistant Remember? Analyzing Conversational Context Recall and Utilization in Voice Interaction Models.
>
> [2] Yongxuan Wu, Runyu Chen, Peiyu Liu, & Hongjin Qian. (2025). LiveLongBench: Tackling Long-Context Understanding for Spoken Texts from Live Streams.

---

> ### Author Response · Authors · 2025-11-27
> **Looking forward to your feedback!**
>
> Dear Reviewer ThYP,
>
> Thank you once again for your insightful feedback. We have conducted additional experiments and revised the draft based on your suggestions. As the discussion phase is drawing to a close, we would like to know if our responses have addressed your concerns. We look forward to hearing from you.
>
> Best regards, Authors

---

### Official Review · Reviewer_2Vzi · 2025-10-26

**Soundness:** 3
**Presentation:** 3
**Contribution:** 2
**Rating:** 4
**Confidence:** 4

**Summary:**

This work focuses on the robustness of voice assistants in third-party interruption (TPI) scenarios, proposing:

1. The TPI-Corpus (containing approximately 80,000 training samples covering 26 interruption scenarios) and the TPI-Bench evaluation set (comprising two parts: TPI-Test for real dual-speaker interruptions, and Janus-Test for text-identical but single-speaker re-synthesized interruptions to expose text shortcut issues)

2. Evaluation across two dimensions: RSF (Response Strategy Following, indicating adherence to processing strategies) and OH (Overall Helpfulness).

3. This work builds upon the Qwen2.5-Omni-7B foundation, progressively incorporating VoiceAssistant-400K and 8k “hard-negative” single-speaker samples to suppress text shortcuts.

Results show that without significantly compromising general speech task capabilities (evaluated using VoiceBench), the model better follows strategies on the TPI-Test and is less misled on the Janus-Test. Additionally, human preference experiments reveal model outputs align closely with reference answer preferences, corroborating the effectiveness of the strategy and training approach.

**Strengths:**

- TPI represents a key obstacle to enhancing the practicality of voice assistants in real-world settings. This paper accurately identifies and systematically addresses this issue, demonstrating significant practical relevance and research value.
- The ingenious design of the Janus-Test stands as one of the most prominent highlights of this paper. By controlling for consistent textual content while varying acoustic features, it provides a direct and powerful tool for detecting shortcut learning issues. This significantly enhances the persuasiveness of evaluation conclusions, demonstrating that the model possesses auditory capabilities.
- The experimental section is comprehensively constructed. Table 1 (e.g., Qwen-it performs well on the TPI-Test but fails on the Janus-Test) reveals the essence of the TPI problem, clearly demonstrating the contribution of each training data component to model performance, and strongly supports the core argument.

**Weaknesses:**

- The entire TPI-Corpus is constructed using text generated by LLM and speech synthesized by TTS systems. The synthesized speech may not fully capture the complex prosodic, emotional, and temporal dynamics present in real human interruptions (e.g., the urgency in the interrupter's voice, the precise timing of overlapping segments, etc.). Supplementing the corpus with a small-scale TPI test set recorded by real humans would further enhance the reliability of the conclusions.
- Simply categorizing interruptions into “actionable” and ‘ignorable’ types is an effective simplification, but more complex scenarios may exist in practice. For instance, in “ignorable” situations, highly intelligent assistants might choose to acknowledge the interruption during their response (e.g., prompting the speaker to continue or confirming whether the interruption should be ignored) rather than completely disregarding it. The binary nature of the current framework may limit exploration of more granular interaction strategies.

**Questions:**

- In both dataset construction (e.g., scenario dialogue generation) and final evaluation, this work employed Qwen3-235B. Could this introduce bias in the LLM's evaluation of models trained on this dataset, potentially leading to inaccurate experimental results? Could the authors provide evaluation results using other LLMs as judge models, or demonstrate consistency between LLM and human evaluations on smaller datasets?
- The current approach to handling “ignorable” interrupts is to completely disregard them. Have you considered alternative, more nuanced handling strategies, such as providing a brief acknowledgment or pause before responding to the primary user? What considerations led to the decision to ignore them entirely (e.g., simplifying the task, or deeming this the optimal strategy)?
- Does the fine-tuned model still produce similar results on the TPI-Test when using speech data from other languages (non-English), non-LibriSpeech domain-specific speakers, and non-Chatterbox domain-specific TTS models?

---

> ### Author Response · Authors · 2025-11-21
> **Author Response (1/n)**
>
> Thank you for your valuable and constructive feedback. Please find my detailed responses to each of your comments below.
> ***
> **[W1] Supplementing the corpus with a small-scale TPI test set recorded by real humans would further enhance the reliability of the conclusions.**
>
> To further validate the practical utility of our TPI-Corpus and ensure our model generalizes well to real-world scenarios, we constructed a new "Real-world TPI Benchmark" consisting of 100 high-quality audio samples. We sourced 3 types of dataset for diversity in acoustic environments and conversational contexts. **1. AMI Meeting Corpus [2]:** Represents real-world meeting scenarios with varying microphone distances (approx. 100 hours). **2. Friends-MMC [1]:** A sitcom dataset (10 seasons, 220 episodes) characterized by multi-party dialogues and frequent interruptions. **3. Human Recordings:** Manual human recordings to mimic daily Voice Assistant usage. The results and analyses are described below and real interaction audio samples are uploaded on the demo page.
>
> **Analysis.**
> Table 1 highlights that the improvements achieved by our model extend consistently from synthetic to real-world settings. On the synthetic benchmark, Qwen2.5-Omni-7B-it-va-hn(our model) delivers substantial gains over the baseline, improving RSF from 0.24 → 0.83 (+0.59) and OH from 3.22 → 4.16 (+0.94). Crucially, these gains persist even under real-world acoustic conditions: compared to the baseline real-world performance (RSF 0.17, OH 3.21), our model achieves RSF 0.60 (+0.43) and OH 4.25 (+1.04).
> These results show two key findings: (1) real-world performance is consistently higher than the baseline by a large margin, and (2) the magnitude of improvement is comparable to the gains observed in the synthetic setting. This directly addresses concerns that the model may overfit to synthetic patterns—the improvements are not only preserved but remain robust across diverse real-world environments, speakers, and noise conditions.
>
> Overall, the strong baseline-relative gains in both synthetic and real-world evaluations validate that models trained on the TPI-Corpus generalize effectively and provide practical utility beyond controlled synthetic scenarios.
>
> **Table 1. Results on high quality real world TPI benchmark**
> | Model                       | TPI-RSF↑ | TPI-OH↑  |
> |-----------------------------|------|------|
> | Qwen2.5-Omni-7B (syn)    | 0.24 | 3.22 |
> | Qwen2.5-Omni-7B-it-va-hn(syn)    | 0.83 | 4.16 |
> | Qwen2.5-Omni-7B (real)    | 0.17 | 3.21 |
> | Qwen2.5-Omni-7B-it-va-hn(real)    | 0.60 | 4.25 |
>
> **[Detailed Benchmark Curation Process]**
>
> **Curation Process:** We employed a two-stage filtering process to identify "Voice Assistant (VA) interaction scenarios interrupted by a third party." First, we used a Qwen-235B-Reasoning model (Qwen-235B) to filter dialogues where a user speaks to a VA followed immediately by an interruption with 5 scale likert rubric score. Second, the authors manually verified all filtered scripts to ensure logical consistency and realistic occurrence in terms of voice assistant scenarios.
>
>     Observation: Despite the vast size of AMI and Friends-MMC, the filtering ratio(LLM, human both) was extremely high and finding natural instances of "interrupted VA interactions" in any multi-party conversation was extremely challenging. We believe this scarcity demonstrates the significant value and necessity of our synthetic TPI-Corpus for training robust models.
> **Environmental Diversity (Human Recordings):** For the human-recorded portion, we reenacted scripts from TPI-Test to test real world acoustic robustness. We selected three distinct environments to introduce varied noise profiles: a room with reverberation, a hallway, and an outdoor setting with background noise. Also we gathered 4 people to record. And they change the role for each script for diversity. The audio samples used for this benchmark and the corresponding inference results are available on our project demo page.

---

> ### Author Response · Authors · 2025-11-21
> **Author Response (2/n)**
>
> **[W2, Q2] The binary nature of the current framework may limit exploration of more granular interaction strategies.**
>
> We appreciate the reviewer’s insightful comment and fully agree that, in real-world settings, a rich spectrum of interaction strategies may exist beyond the two categories discussed in the paper. Indeed, highly capable assistants might choose to acknowledge an interruption, briefly pause, or ask for clarification rather than fully ignoring it. We recognize that such nuanced conversational behaviors can vary across individuals, cultures, personalities, and usage contexts.
>
> In our work, however, the goal is to propose a general and flexible framework rather than to prescribe a single “correct” conversational policy. As defined in Line 144, we classify an interruption as ignorable only when the assistant completely disregards the third-party speech. All other behaviors(a complementary set)—including acknowledgments, confirmations, partial pauses, or brief handling of the interruption—fall into the broader category of actionable in our framework . In other words, our binary categorization is not intended to constrain the space of strategies; rather, it provides a minimal conceptual foundation upon which many nuanced actionable behaviors can be organized. As we illustrated example strategies in Appendix D, the other strategies similar to your suggestion(e.g. seeking confirmation, temporarily maintaining silence) could be customized as “Actionable” by end-users within our frameworks.
>
> We believe that determining which actionable strategy is “optimal” in a given situation is inherently subjective and deeply dependent on human preferences. Instead of enforcing a single normative reference, our objective is to offer a versatile framework that end-users can adapt and personalize according to their conversational expectations. The reference strategies we demonstrate empirically serve as an example—evidence that the proposed framework can support effective model behavior, not a restriction on the broader space of interaction dynamics.
>
>
> Thus, we respectfully argue that the binary formulation does not limit exploration but instead enables users and future work to build richer, personalized, and context-sensitive interruption-handling strategies on top of a simple, coherent foundation. **We will release the automation pipeline code publicly, and we have revised the draft to reflect this in Abstract.**
>
> ***
> **[Q1] Could the authors provide evaluation results using other LLMs as judge models, or demonstrate consistency between LLM and human evaluations on smaller datasets?**
>
> Thank you for pointing that out. To directly address the reviewer's concern, we conducted a cross-verification using GPT-5-mini and human evaluation for our robustness to evaluation method. In order to check if our evaluation result aligns with human judgement, we further evaluate our model’s response with human judgement using “Amazon Mechanical Turk”. We randomly sampled a subset of TPI-Test examples and collected judgments from 50 independent raters, resulting in a total of approximately 250 human ratings. The interfaces utilized for human evaluation are added in **Appendix E** in the revised draft.
>
> **Table 2** shows that the other evaluation results show similar tendency with the one evaluated with Qwen3-235B. We also calculated the Pearson correlation of each evaluation strategy. **Table 3** demonstrates that evaluations produced by the Qwen3-235B model exhibit strong correlations with results obtained from alternative evaluators, including other LLM-based assessors and human judgments. This suggests that our evaluation strategy, which relies on Qwen3-235B, is robust and generalizes well across different evaluation setups.
>
> **Table 2. Other evaluation results including human evaluation.**
> | Model                       | TPI-RSF↑ | TPI-OH↑  | Janus-RSF↓ | Janus-OH↑  |
> |-----------------------------|------|------|------|------|
> | Qwen2.5-Omni-7B-it-va-hn(eval with Qwen3-235B)    | 0.83 | 4.16 | 0.16 | 4.80 |
> | Qwen2.5-Omni-7B-it-va-hn(gpt-5-mini)    | 0.8 | 4.42 | 0.17 | 4.73 |
> | Qwen2.5-Omni-7B-it-va-hn(human)    | 0.88 | 3.87 | -| - |
>
>
> **Table 3. Pearson Correlation among different evaluation strategies.(p<0.05)**
>
> | Pair                           | TPI-RSF Corr | TPI-OH Corr |Janus-RSF Corr | Janus-OH Corr |
> |--------------------------------|-----|-----|-----|----|
> | Qwen3-235B ↔ GPT-5-mini        |   0.85     |  0.83     |     0.86    |   0.85     |
> | Qwen3-235B ↔ Human             |   0.87      |   0.69     |  -       |   -     |

---

> ### Author Response · Authors · 2025-11-21
> **Author Response (3/3)**
>
> **[Q3] Producing similar results on other languages (non-English), non-LibriSpeech domain-specific speakers, and non-Chatterbox domain-specific TTS models.**
>
>
> We appreciate the reviewer’s question regarding generalization beyond English, MultiLingual-LibriSpeech speakers, and Chatterbox-based TTS. Although our work focuses primarily on English data, we believe the proposed approach generalizes naturally across languages and speaker domains because the core mechanism we rely on—vocal shift as an acoustic cue for interruptions—is language-agnostic. As long as a speaker change produces a distinguishable acoustic difference, the model should transfer to other languages as well. Moreover, since we are planning to release an end-to-end automated pipeline for data generation and evaluation, users will be able to create multilingual TPI datasets and run the full evaluation process in other languages with minimal effort.
>
> To examine whether our model overfits to a specific speaker distribution or TTS backend, we conducted additional evaluations on our Real-world TPI Benchmark, which includes speakers and recording conditions not present in MultiLingual-LibriSpeech or Chatterbox. As noted in Weakness #1 and demonstrated in **Table 1**, the model trained solely on TPI-Corpus generalizes robustly to real-world acoustic settings, suggesting that it learns to detect speaker transitions rather than memorizing speaker identities or TTS-specific artifacts.
>
> Once again, we really appreciate giving us the opportunity to further strengthen our paper.
>
> ***
>
> **References**
>
> [1] Yueqian Wang, Xiaojun Meng, Yuxuan Wang, Jianxin Liang, Qun Liu, & Dongyan Zhao. (2024). Friends-MMC: A Dataset for Multi-modal Multi-party Conversation Understanding.
>
> [2] Carletta, J., Ashby, S., Bourban, S., Flynn, M., Guillemot, M., Hain, T., Kadlec, J., Karaiskos, V., Kraaĳ, W., Kronenthal, M., Lathoud, G., Lincoln, M., Lisowska, A., McCowan, I., Post, W., Reidsma, D., & Wellner, P. (2005). The AMI Meeting Corpus: A Pre-announcement. In Machine Learning for Multimodal Interaction (pp. 30–39). Springer.

---

> > ### Comment · Reviewer_2Vzi · 2025-11-27
> >
> > I thank the authors for the additional experiments and their response, which address my original concerns. Given the new results, I would like to raise my score from 4 to 6, acknowledging the work's potential contribution to the community.

---

> > > ### Author Response · Authors · 2025-11-27
> > > **Thank you for raising the score!**
> > >
> > > Dear Reviewer 2Vzi,
> > >
> > > We're glad to hear that your concerns have been resolved, and we truly appreciate you raising the score. Once again, We sincerely appreciate the effort and valuable feedback you've put into the review process. Wishing you a great day!
> > >
> > > Best,
> > >
> > > Authors

---

### Official Review · Reviewer_Uiy5 · 2025-10-31

**Soundness:** 2
**Presentation:** 3
**Contribution:** 2
**Rating:** 2
**Confidence:** 4

**Summary:**

This work studies the ability of voice assistant systems to handle third-party interruptions using acoustic and textual cues. To support this, the authors construct a large synthetic dataset of interruptions by injecting simulated interjections from third speakers into user-VA dialogues, and propthat integrates audio features and is trained with hard negative examples designed to challenge text-only shortcutting. First, they show that baseline models struggle to handle these synthetic interruptions when speaker identity is manipulated, and that training on synthetic interruptions significantly improves performance on a evaluation set sampled from the synthetic corpus. Then, they introduce a paired contrast set designed to isolate acoustic grounding, and show that hard negative training further improves robustness by encouraging models to rely on speaker change cues rather than transcript semantics alone.

**Strengths:**

- The work introduces a synthetic data pipeline for constructing interruptions which can be used to finetune models to improve their robustness to third party interruptions.
- The work shows that they are able to finetune models on this dataset without significant degradation to other voice assistant capabilities.
- The Janus-Test counterfactual setup is a clever way to probe whether models are actually using speaker change information versus relying on semantic shortcuts.

**Weaknesses:**

The work does not provide any clear separated validation for the effectiveness of their synthetic data pipeline. The paper generates interruptions synthetically and solely validate the utility of finetuning on this data by evaluating on data generated using the same synthetic data generation process. This leaves me significantly unclear whether this dataset helps with real world interruptions.

While the synthetic data process seems likely valuable for training, I would expect to see one of the following things:
(1) TPI-Test could be validated by heavily quality filtering using human judgements to confirm that this subset is one that humans agree is high quality and realistic of real world interruptions.
(2) TPI-Corpus (train split) could be validated using a relatively small sample of real human interruptions gathered from the web (for example, filtered from large web-scale audio corporate such as YODAS2).

Real world interruptions involve far more auditory cues than simply changing speaker voice (some speakers may be closer or further away from the microphone for example) and it seems necessary to do at least some amount of validation of the data using either human judgements on the proposed test data or additional human evaluation on the trained models. Otherwise, I worry that this dataset could only be improving the ability of the model to respond to the synthetic data distribution.

**Questions:**

- What is just Qwen2.5-Omni-it? The other models seem to be defined, but this one isn't defined within the paper. https://huggingface.co/collections/Qwen/qwen25-omni doesn't show this being one of the release Qwen 2.5 Omni models, but the lack of definition seems odd if this is newly defined in this paper.

- Was there any form of non-synthetic validation done of the data generation process? Given that the core argument is that this should help the model deal with real auditory

- Is there any form of synthetic variation for the TPI's created other than synthetic speaker swaps? It seems possible to add other forms of realistic noise induced by synthetic interruptions.

- Are all synthetic interruptions strictly turn based (as shown in the figures)? Real world interruptions seem likely to have overlapping speech, but this isn't mentioned in the work.

- Do the LLM-based helpfulness judgments correlate with human judgments in any way? Were any human evaluations of the model responses collected to ground those metrics?

---

> ### Author Response · Authors · 2025-11-21
> **Author Response (1/n)**
>
> Thank you for your valuable and constructive feedback. Please find my detailed responses to each of your comments below.
> ***
> **[W1] Real-world Utility and Generalization**
>
> To further validate the practical utility of our TPI-Corpus and ensure our model generalizes well to real-world scenarios, we constructed a new "Real-world TPI Benchmark" consisting of 100 high-quality audio samples. We sourced 3 types of dataset for diversity in acoustic environments and conversational contexts. **1. AMI Meeting Corpus [2]:** Represents real-world meeting scenarios with varying microphone distances (approx. 100 hours). **2. Friends-MMC [1]:** A sitcom dataset (10 seasons, 220 episodes) characterized by multi-party dialogues and frequent interruptions. **3. Human Recordings:** Manual human recordings to mimic daily Voice Assistant usage. The results and analyses are described below and real interaction audio samples are uploaded on the demo page.
>
> **Analysis.**
> Table 1 highlights that the improvements achieved by our model extend consistently from synthetic to real-world settings. On the synthetic benchmark, Qwen2.5-Omni-7B-it-va-hn(our model) delivers substantial gains over the baseline, improving RSF from 0.24 → 0.83 (+0.59) and OH from 3.22 → 4.16 (+0.94). Crucially, these gains persist even under real-world acoustic conditions: compared to the baseline real-world performance (RSF 0.17, OH 3.21), our model achieves RSF 0.60 (+0.43) and OH 4.25 (+1.04).
> These results show two key findings: (1) real-world performance is consistently higher than the baseline by a large margin, and (2) the magnitude of improvement is comparable to the gains observed in the synthetic setting. This directly addresses concerns that the model may overfit to synthetic patterns—the improvements are not only preserved but remain robust across diverse real-world environments, speakers, and noise conditions.
>
> Overall, the strong baseline-relative gains in both synthetic and real-world evaluations validate that models trained on the TPI-Corpus generalize effectively and provide practical utility beyond controlled synthetic scenarios.
>
> **Table 1. Results on high quality real world TPI benchmark**
> | Model                       | TPI-RSF↑ | TPI-OH↑  |
> |-----------------------------|------|------|
> | Qwen2.5-Omni-7B (syn)    | 0.24 | 3.22 |
> | Qwen2.5-Omni-7B-it-va-hn(syn)    | 0.83 | 4.16 |
> | Qwen2.5-Omni-7B (real)    | 0.17 | 3.21 |
> | Qwen2.5-Omni-7B-it-va-hn(real)    | 0.60 | 4.25 |
>
> **[Detailed Benchmark Curation Process]**
>
> **Curation Process:** We employed a two-stage filtering process to identify "Voice Assistant (VA) interaction scenarios interrupted by a third party." First, we used a Qwen-235B-Reasoning model (Qwen-235B) to filter dialogues where a user speaks to a VA followed immediately by an interruption with 5 scale likert rubric score. Second, the authors manually verified all filtered scripts to ensure logical consistency and realistic occurrence in terms of voice assistant scenarios.
>
>     Observation: Despite the vast size of AMI and Friends-MMC, the filtering ratio(LLM, human both) was extremely high and finding natural instances of "interrupted VA interactions" in any multi-party conversation was extremely challenging. We believe this scarcity demonstrates the significant value and necessity of our synthetic TPI-Corpus for training robust models.
> **Environmental Diversity (Human Recordings):** For the human-recorded portion, we reenacted scripts from TPI-Test to test real world acoustic robustness. We selected three distinct environments to introduce varied noise profiles: a room with reverberation, a hallway, and an outdoor setting with background noise. Also we gathered 4 people to record. And they change the role for each script for diversity. The audio samples used for this benchmark and the corresponding inference results are available on our project demo page.
>
> ***
>
> **[W2] Evaluation of the Synthetic Generation Pipeline**
> To address the concern regarding the realism of our synthetic generation pipeline, we conducted a human evaluation to verify whether the generated samples reflect real-world conversational dynamics.
> We utilized Amazon Mechanical Turk (MTurk) for the study. We randomly sampled the subset of the TPI-Test and collected judgments from 50 independent raters, resulting in a total of approximately 500 human ratings.
> Participants rated the audio samples on a 3-point Likert scale, where 3 indicates "Realistic", 2 indicates “Moderate” and 1 indicates "Unrealistic."
>
> The evaluation yielded an average score of 2.87(Table 2), demonstrating a high degree of perceived realism.
> The detailed interface used for the MTurk study have been added to **Appendix E** of the revised paper for reproducibility.
>
> **Table 2. Human evaluation on naturalness of TPI-Test samples (mean ± 95% CI)**
> | Dataset                       | naturalness(3 scale)  |
> |-----------------|------------|
> | TPI-Test      | 2.87 ± 0.034 |

---

> > ### Comment · Reviewer_Uiy5 · 2025-11-25
> > **Response to (1/n)**
> >
> > ```
> > First, we used a Qwen-235B-Reasoning model (Qwen-235B) to filter dialogues where a user speaks to a VA followed immediately by an interruption with 5 scale likert rubric score. Second, the authors manually verified all filtered scripts to ensure logical consistency and realistic occurrence in terms of voice assistant scenarios.
> > ```
> >
> > It is not clear how this filtering step yields any results if the rubric is accurately followed. Neither the sitcoms in that dataset nor the AMI corpus involve a voice assistant in the interactions. Which party is being treated as the voice assistant in these dialogues? Likewise, which speaker is treated as the user? Is the ground truth response still generated from a language model given the preferred strategy or is the ground truth response taken from the actual interaction data within AMI/Friends?
> >
> > Simulating scripts from TPI-test is a welcome addition, but doesn't fully address the synthetic data concerns! The response distribution is still from the same model which the test set is generated from so improvements still can simply come from matching this models generative distribution more consistently.
> >
> > ```
> > To address the concern regarding the realism of our synthetic generation pipeline, we conducted a human evaluation to verify whether the generated samples reflect real-world conversational dynamics.
> > ```
> >
> > However, the instructions in the appendix indicate that this was a **text-only** verification. This does not address my concern, which is specifically about whether the **synthetic audio** generations are realistic.
> >
> > From Appendix E:
> >
> > > “You will read a short but various scenario where a Primary User talks to Voice Assistant, and a Third Party interrupts the primary user for any purpose.”
> >
> > As a minor point: the significance testing in your table appears to make incorrect assumptions. The reported confidence interval for a 3-point scale is:
> >
> > ```
> > Dataset       naturalness (3 scale)
> > TPI-Test      2.87 ± 0.39
> > ```
> >
> > which implies values outside the possible range (e.g., 2.87 + 0.39 = 3.26 > 3). This is a small issue likely from assuming the true score has a normal distribution, but is the type of error that occurs from quickly running non-trivial new experiments in a compressed period!

---

> > > ### Author Response · Authors · 2025-11-26
> > > **Further Response(1/n)**
> > >
> > > Thank you for your quick reply!  We sincerely appreciate your willingness to consider our work and your sharp eye regarding the evaluation details. We understand your concern that the additional experiments might appear "rushed." However, we would like to respectfully assure you that these were conducted with strict rigor to validly address the gap between synthetic and real-world distributions.
> > >
> > > ***
> > > **[Q1] It is not clear how this filtering step yields any results if the rubric is accurately followed. Neither the sitcoms in that dataset nor the AMI corpus involve a voice assistant in the interactions. Which party is being treated as the voice assistant in these dialogues? Likewise, which speaker is treated as the user?**
> > >
> > > Sorry for omitting the details of the filtering process. We clarify the rigorous filtering process for the real-world benchmark below.
> > >
> > > **[Why we choose multi party dialogue?]**
> > >
> > > To the best of our knowledge, a "Interrupted Voice Assistant" dataset does not currently exist. Therefore, our goal was to filter real-world conversational moments that can be luckily interpreted as our target scenario: a primary user speaks to Voice Assistant, and a third party interrupts before it  responds. Given this setup—that dialogues involving more than three speakers are more likely to yield more data after the filtering process—we utilized multi-party conversation datasets, AMI (100h) and Friends (70h).
> > >
> > > **[Filtering Criterion]**
> > >
> > > First, we grouped all consecutive utterances by two different speakers from the transcripts. Following our task definition (Section 2), we treated the first speaker as the "Primary User" and the second utterance as the "Third-Party Interrupter." We then filtered for samples satisfying three strict criteria:
> > >
> > > **Criterion 1:** The "Primary User's" utterance must be plausible as a command or query directed at a Voice Assistant. We selected utterances that could be interpreted as addressing a VA.
> > >
> > > **Criterion 2:** The second speaker’s utterance (Third Party) must be interpretable as an interruption. We filtered for turns where semantic content of the second speaker could be regarded as a third-party interruption.
> > >
> > > **Criterion 3:** The "Primary User's" request must be answerable by a text-based SLM(LLM). We excluded requests requiring infeasible actions such as physical actions or visual grounding (e.g., "Open the window," "Turn up the volume," "Look at this", “Search for a specific word in Google” etc.).
> > >
> > >       Findings: We found that when presented with infeasible requests, the model often outputs a   safety response (e.g., “I cannot perform that(e.g. physical) actions due to the nature of the text-based-system model”). This makes the model fail not because it mishandled the interruption, but because of its modality constraints. Excluding these ensures the evaluation focuses purely on the model's ability to handle interruptions.
> > >
> > > **[Filtering Process]**
> > >
> > > We applied a rigorous two-stage filtering pipeline (LLM-based pre-filtering followed by manual verification).
> > >
> > > **Stage 1:** Given the vast volume of the source audio, manual filtering the whole dataset was prohibitively expensive in terms of both time and cost. Therefore, we employed a reasoning model(Qwen3-235B-2507-reasoning) as a preliminary filter to identify samples satisfying the three criteria that were rated 5. The filtering rejection rate was extremely high (>99%). This scarcity indicates that these specific interruption scenarios are rare in general multi-party dialogues, thereby highlighting the unique value of our curated dataset. The LLM prompt for this stage has been added in Appendix F(Last page of revised draft).
> > >
> > > **Stage 2:** All authors participated in a second round of manual filtering process for the samples selected by the LLM. We listened to every audio sample to ensure they acoustically met the criteria. Approximately 70% of the candidate samples were further discarded during this phase. Samples were primarily rejected if they did not acoustically resemble a genuine interruption (e.g., a long pause existed before the "interruption") or if the scenario was contextually unnatural (e.g., the Primary User explicitly used pronouns or names addressing a human interlocutor, which breaks the assumption of addressing a VA) or annotation errors.

---

> ### Author Response · Authors · 2025-11-21
> **Author Response (2/n)**
>
> **[Q1] What is just Qwen2.5-Omni-it?**
>
> We apologize for the confusion caused by the naming.
> “-it” refers to the model fine-tuned only on the interruption data split of our TPI-Corpus (i.e., the model trained exclusively on the interruption-specific supervision). It is not a released model from the official Qwen2.5-Omni series, but rather a variant defined within our work.
>
> We also acknowledge that the abbreviation “it” may be unclear and could easily be mistaken for a standard model name. To prevent similar confusion for other readers and reviewers, we will revise the notation after the rebuttal phase—specifically, we will rename Qwen2.5-Omni-it → TPI right before the end of rebuttal phase to prevent misunderstanding from other reviewers within rebuttal phase.
>
> ***
>
> **[Q2] any form of non-synthetic validation done of the data generation process?**
>
> We acknowledge that conducting full-scale human validation on all synthetic data is challenging due to the size of the corpus. Following your suggestion, we performed targeted human validation on TPI-Test, confirming that the generated scenarios are realistic and well-aligned with how humans perceive Voice Assistant interaction patterns.
> Moreover, to directly assess whether models trained on our synthetic dataset generalize to real auditory conditions, we constructed a Real-world TPI Benchmark using 100 high-quality audio samples from AMI, Friends-MMC, and human-recorded environments.
> As shown in Table 1, models trained solely on TPI-Corpus maintain strong performance on this real world benchmark, demonstrating that our synthetic data generation process transfers effectively to real-world acoustic settings.
>
> ***
>
> **[Q3] any form of synthetic variation for the TPI's created other than synthetic speaker swaps?**
>
> We did not explicitly add additional synthetic noise beyond speaker substitutions. However, our synthetic audio is generated using TTS-based voice cloning, where each script is rendered using real-world recordings from the reference audios. In practice, we observed that the Chatterbox TTS system does not only reproduce the speaker’s vocal characteristics but also tends to implicitly carry over environmental attributes present in the real world recorded audio (e.g., mild white noise, reverberation, microphone artifacts). We uploaded conversion examples on the demo page. As a result, the generated TPI samples naturally contain not only clear acoustic conditions but also a diverse set of acoustic conditions without requiring us to manually inject noise.
>
> Importantly, the effectiveness of this design choice is supported by our Real-world TPI Benchmark results. Despite being trained only on synthetic TPI-Corpus, the model generalizes well to real audio recorded across different environments. This suggests that the normal noise characteristics implicitly captured through TTS voice cloning, combined with the structural variation in our synthetic scripts, are already sufficient for achieving real-world transfer.

---

> ### Author Response · Authors · 2025-11-21
> **Author Response (3/3)**
>
> **[Q4] Are all synthetic interruptions strictly turn based?**
>
> Our synthetic interruptions are primarily turn-based, but they are not strictly disjoint. As described in Section 3.4, we introduce controlled temporal overlap between the Voice Assistant (VA) utterance and the third-party interruption by sampling the overlap duration from a Gaussian distribution. This creates natural partial overlap while keeping the interruption timing realistic and stable for model training.
>
> We acknowledge that real-world conversational overlap can be far more diverse (e.g., full overlap, intermittent overlap, partial overlap, staggered interruptions). However, current spoken language models (SLMs) still struggle significantly with even moderate overlapping speech. Incorporating the full space of complex overlap patterns would dramatically expand the scope of the work and obscure our central focus: teaching models to detect the vocal-shift cue that signals a third-party interruption.
> Indeed, a representative failure case(details in Appendix A) illustrates this challenge. When even a small portion of overlap occurs near a phonetic boundary, models such as GPT-4o-audio-preview often misrecognize the corrupted segment and produce nonsensical answers. For example:
>
> **Failure Case (GPT-4o-audio-preview)**
>
>      Primary User: Rewrite the sentence to show the definition of the word manifold.
>      Third Party: Hey can you check if we need more time on the laundry before starting dinner?
>      Model Answer: Here’s the sentence with the definition of “man”:“Hey, can you check if we need more time on the laundry before starting dinner?” (where ‘man’ means to take charge of or manage a task or responsibility).”
>
> These types of errors show that even state-of-the-art SLMs are not yet robust to highly overlapped speech. For this reason, we intentionally limit the variability of overlap during data generation to avoid confounding the core phenomenon.
> That said, synthetic generation offers a unique advantage: since we provide separated utterances as well, our pipeline can be easily extended to curate a broad range of complex overlap patterns. We view this as an exciting direction for future work, and our synthetic approach and dataset provides a strong foundation for systematically studying overlap-heavy conversational scenarios.
>
> ***
>
> **[Q5] Do the LLM-based helpfulness judgments correlate with human judgments in any way?**
>
> Thank you for pointing that out. To directly address the reviewer's concern, we conducted a cross-verification using GPT-5-mini and human evaluation for our robustness to evaluation method. In order to check if our evaluation result aligns with human judgement, we further evaluate our model’s response with human judgement using “Amazon Mechanical Turk”. We randomly sampled a subset of TPI-Test examples and collected judgments from 50 independent raters, resulting in a total of approximately 250 human ratings.The interfaces utilized for human evaluation are added in **Appendix E** in the revised draft.
>
> **Table3** shows that the other evaluation results show similar tendency with the one evaluated with Qwen3-235B. We also calculated the Pearson correlation of each evaluation strategy. **Table 4** demonstrates that evaluations produced by the Qwen3-235B model exhibit strong correlations with results obtained from alternative evaluators, including other LLM-based assessors and human judgments. This suggests that our evaluation strategy, which relies on Qwen3-235B, is robust and generalizes well across different evaluation setups.
>
> **Table 3. Other evaluation results including human evaluation**
> | Model                       | TPI-RSF↑ | TPI-OH↑  | Janus-RSF↓ | Janus-OH↑  |
> |-------|------|-----|------|-----|
> | Qwen2.5-Omni-7B-it-va-hn(eval with Qwen3-235B)    | 0.83 | 4.16 | 0.16 | 4.80 |
> | Qwen2.5-Omni-7B-it-va-hn(gpt-5-mini)    | 0.8 | 4.42 | 0.17 | 4.73 |
> | Qwen2.5-Omni-7B-it-va-hn(human)    | 0.88 | 3.87 | -| - |
>
> **Table 4. Pearson Correlation among different evaluation strategies. (p<0.05)**
> | Pair                           | TPI-RSF Corr | TPI-OH Corr | Janus-RSF Corr | Janus-OH Corr |
> |------|----|----|-----|----|
> | Qwen3-235B ↔ GPT-5-mini        |   0.85     |  0.83     |     0.86    |   0.85     |
> | Qwen3-235B ↔ Human             |   0.87      |   0.69     |  -       |   -     |
>
>
> Once again, we really appreciate giving us the opportunity to further strengthen our paper.
>
> ***
>
> **References**
>
> [1] Yueqian Wang, Xiaojun Meng, Yuxuan Wang, Jianxin Liang, Qun Liu, & Dongyan Zhao. (2024). Friends-MMC: A Dataset for Multi-modal Multi-party Conversation Understanding.
>
> [2] Carletta, J., Ashby, S., Bourban, S., Flynn, M., Guillemot, M., Hain, T., Kadlec, J., Karaiskos, V., Kraaĳ, W., Kronenthal, M., Lathoud, G., Lincoln, M., Lisowska, A., McCowan, I., Post, W., Reidsma, D., & Wellner, P. (2005). The AMI Meeting Corpus: A Pre-announcement. In Machine Learning for Multimodal Interaction (pp. 30–39). Springer.

---

> > ### Comment · Reviewer_Uiy5 · 2025-11-25
> > **Overall Reply to Rebuttal**
> >
> > I appreciate the effort the authors have shown adding additional experiments and validations of the utility of their training data. I have adjusted my score from a 2 to a 4 accordingly.
> >
> > Ultimately, I still think this work would benefit from a  more careful (and less rushed) revision process to carefully validate the synthetic data with a well designed (even if a small set) of manual data which *does not involve the synthetic data generation procedure in the loop*.
> >
> > The contribution of datasets in which the data is synthetically generated, the test data is synthetically generated, and the underlying metrics are produced by LLM as a judge is, to me, a relatively incremental contribution without accompanying careful manual validation showing utility in the real world. While I do appreciate the new experiments in this direction (and have adjusted my score accordingly), these experiments have several signs (easy to catch errors, odd mismatches with the use case, and continuing reliance on parts of the synthetic data pipeline) that indicate they were done for expediency for rebuttal rather than as carefully done validation of the work. Personally, I feel the work is not above the bar for acceptance for a dataset work, but if another reviewer were a strong proponent for acceptance I would not vehemently argue against it with the additional experiments.

---

> ### Author Response · Authors · 2025-11-26
> **Further Response(2/n)**
>
> **[Q2]  Is the ground truth response still generated from a language model given the preferred strategy or is the ground truth response taken from the actual interaction data within AMI/Friends?**
>
> To clarify, there is no "Ground Truth Voice Assistant response” derived from the AMI or Friends datasets. As a kind reminder, our evaluation framework also evaluates, “without ground truth answer”, whether the model adheres to the designated strategies **(RSF)** and whether it responds correctly without misinterpreting the interruption context **(OH)**. Instead we have ground truth criteria (if it is actionable class or not **(RSF)**, and 5 scale detailed rubric **(OH)**) and Actionable and Ignorable classes labeled according to designated strategies. (Actionable label of real world benchmark samples are also labeled with the exactly same prompt used for TPI-corpus labeling before)
>
> In the example below, sampled from the Friends part, only the utterances of the Primary User and the Interrupter exist as a sample of real world benchmark; there is no ground truth answer for the assistant. We query our model with this audio segment and measure its response based on its strategic decision (**(RSF)**, checking if the model follows predefined answer strategies description) and the appropriateness of its generated response (**(OH)**), checking whether the model misinterprets the audio segment as a single or two speakers, thereby generating totally awkward answer).
>
> **Example.** It could be interpreted as a primary user seeking consultation from a voice assistant, but then a third party suddenly intervenes.”
>
>       Primary User: "Oh, Monica and Chandler's recommendation. I want it to sound smart but.. I don't know any big words or anything, so..."
>       TPI (Third Party): "Why don't you use your Thesaurus?"
>
> ***
>
> **[Q3]  The response distribution is still from the same model which the test set is generated from so improvements still can simply come from matching this models generative distribution more consistently.**
>
> The primary objective of “Simulating scripts from TPI-test” was to address concerns regarding the acoustic domain gap—specifically, whether a model trained on synthetic voices would generalize well to real-speaker acoustic settings. By keeping the transcripts (semantic variable) constant and varying only the audio(syn->real), we aimed to demonstrate that our model still performs robustly when transitioning from synthetic to real-speaker settings, in line with your concern about its transferability to real-speaker acoustic conditions. Plus, to further address the various response distributions, we intended to include cases from both AMI and Friends (whose transcripts are from different distributions of  the training).
>
> In summary**[Q1,2,3]**, we established a benchmark based on rigorous criteria. The demonstrated high performance on this real-world benchmark validates the effectiveness of our synthetic data strategy, confirming that the model successfully leverages acoustic voice transitions to robustly handle interruptions.
>
> ***
>
> **[Q4] As a minor point: the significance testing in your table appears to make incorrect assumptions**
>
> We sincerely apologize for the confusion caused by the notation in our previous response. We would like to clarify that the statistical analysis itself was conducted rigorously. However, in the process of simultaneously addressing comprehensive feedback from four reviewers, we made a clerical error in transcribing the results. We mistakenly presented the standard deviation instead of the corresponding confidence interval, which led to the ambiguity you rightly pointed out.
>
> However, we respectfully clarify that this was strictly an inadvertent notation mistake, not an error in calculation. The statistical details are as follow:
>
> **Mean Score:** 2.87
>
> **Standard Deviation:** 0.39 **(The value of std remains same)**
>
> **Margin of Error:** 1.96 x 0.39 / sqrt(500) = 0.034
>
> **95% Confidence Interval:** [2.836, 2.904]
>
> The reported values indicate that our samples are not widely scattered but are densely clustered at the high end of the quality spectrum. The tight interval further confirms that the estimation of the mean is highly stable. Thus, these statistics should be interpreted as a strong indicator of our model's consistently high naturalness.
>
> We edited the CI value of Table2 in "Author Response (1/n)."

---

> ### Author Response · Authors · 2025-11-26
> **Further Response(3/3)**
>
> **[Q5] However, the instructions in the appendix indicate that this was a text-only verification. This does not address my concern, which is specifically about whether the synthetic audio generations are realistic.**
>
> You rightly pointed out that our previous Appendix description implied a text-only evaluation. We previously misunderstood you are concerned about the textual scenarios itself. To rectify this and fully address the reviewer's concern, we have conducted a “comprehensive&heavy” audio-text-grounded human evaluation. We randomly sampled the subset of the TPI-Test and collected a total 2000 human ratings across 200 independent raters, matching the total number of human ratings to the number of samples in the TPI-Test (2K). We asked evaluators to assess the samples based on the following prompt:
>
>       "Please judge: Does this interruption sound like a real human conversation? Decide whether this audio clip represents an interruption scenario that sounds both Realistic (Could happen in real life) and Natural (e.g., interruption timing, tone, etc.)."
> (The specific interface details have been added to Appendix E.)
>
> **[Results]**
>
> **Mean Score:** 2.63
>
> **Standard Deviation:** 0.5084.
>
> **Margin of Error:** 1.96 * 0.5084 / sqrt(2000) = 0.0223
>
> **95% Confidence Interval (CI):** [2.608, 2.652]
>
> **Table 2. Human evaluation on comprehensive naturalness(audio+text) of TPI-Test samples (mean ± 95% CI)**
> | Model                       | comprehensive realism(3 scale)  |
> |-----------------------------|------------|
> | TPI-Test      | 2.63 ± 0.0223 |
>
>
>
> Ideally, Table 2 shows that the audio evaluation (2.63) aligns with the scenario realism (2.87), demonstrating that our synthesis pipeline successfully preserved the conversational naturalness. Achieving a score of 2.63/3.0 (approx. 88% of the perfect score) is particularly significant given the complexity of modeling interruption—a task for which no “interruption-specialized” off-the-shelf TTS currently exists. This confirms that our pipeline effectively bridges the gap between semantic plausibility and acoustic realism.
>
> Along with **[Q4]**, we really appreciate the **reviewer MWRj** for giving us the opportunity to correct and further strengthen our arguments.
>
> ***
> Overall, we understand that the omission of specific details for simplicity in the initial response may have unintentionally created the impression that our experiments were conducted hastily. However, we would like to respectfully clarify that all experiments were executed with strict methodological rigor, as evidenced by the detailed protocols shared above. Also we believe  that all misunderstandings are handled according to our additional response as follow.
>
> - easy to catch errors **[Q4,5]**
> - odd mismatches with the use case **[Q1,2,3]**
> - continuing reliance on parts of the synthetic data pipeline **[Q2,3]**
>
> Furthermore, we reflected the all related experiment results in a revised draft with red text. We are happy that our dataset and arguments have been substantially strengthened through this discussion with you. We hope our additional details fully addresses your concerns; if any ambiguity remains, please let us know to clarify it.

---

### Official Review · Reviewer_zKni · 2025-11-01

**Soundness:** 3
**Presentation:** 3
**Contribution:** 3
**Rating:** 4
**Confidence:** 3

**Summary:**

This paper presents a systematic study of voice assistants’ performance in third-party interruption scenarios, introducing the first large-scale TPI-aware dataset and a comprehensive evaluation framework. The work is innovative in dataset construction, task definition, and evaluation methodology, and it can significantly advance the practical deployment of voice assistants in complex multi-speaker environments. The overall structure is clear, experiments are thorough, and the paper offers substantial academic and practical value.

**Strengths:**

1. The dataset is very useful and provides a solid foundation for research on third-party interruptions in voice assistants.
2. The experiments are comprehensive and effectively demonstrate the method’s effectiveness.
3. The paper is well-structured and clearly written

**Weaknesses:**

1. The evaluation is not comprehensive enough, as results for some state-of-the-art closed-source dialogue models (such as GPT-4o-audio, Gemini 2.5 Pro, etc.) are missing.
2. The case presentation is somewhat insufficient; while the paper showcases some cases from the dataset, it lacks output cases from comparative models, making it difficult to intuitively understand the differences in metrics.
3. Although the paper proposes a valuable dataset and evaluation strategy, there is a lack of methodological innovation in model training.

**Questions:**

1. In constructing the TPI-Corpus, how do you ensure diversity across different scenarios and speakers?
2. User preferences for handling interruptions may vary. What directions for improvement do you see for your work in this aspect?
3. In Section 3.4 you mention that the open-source LLM’s performance is close to closed-source models. Is there a quantified comparison for this task? Also, how large is the gap between LLM-generated and human-generated results?

---

> ### Author Response · Authors · 2025-11-21
> **Author Response (1/n)**
>
> Thank you for your valuable and constructive feedback. Please find my detailed responses to each of your comments below.
> ***
> **[W1] The lack of closed-source SOTA model results.**
>
> Thank you for pointing this out. We agree that comparing closed-source models is crucial to bridge the gap between industrial and academic landscapes. Before discussing the results, we briefly remind the metrics used: RSF (Response Strategy Following) measures whether the model strictly adheres to the predefined interruption-handling strategy (e.g., ignoring the interrupter vs. attending to the user), whereas OH (Overall Helpfulness) evaluates the naturalness and semantic quality of the response on a 5-point Likert scale.
>
> Following your suggestion, we conducted additional experiments using one of the current state-of-the-art closed models, GPT-4o-audio-preview. Using both the TPI-Corpus and Janus-Test benchmarks, we observed that although the closed-source model exhibits certain improvements in OH compared to existing open-source baselines, this increase primarily reflects better overall response quality rather than an enhanced ability to handle speaker changes. We consistently found clear failure cases indicating that the model still struggles to differentiate between speakers based on acoustic cues, and its performance remains lower than ours. Moreover, the auxiliary metrics (BLEU and ROUGE-L) show relatively high values, which suggests that the model produces nearly identical responses even when the speaker configuration changes. This further supports our claim that recent spoken language models rely mainly on textual information and fail to incorporate speaker-specific vocal characteristics.
>
> **Table1. Performance comparison between open-source model and closed-source model**
> | Model                       | TPI-RSF↑ | TPI-OH↑  | Janus-RSF↓ | Janus-OH↑  | BLEU↓ | ROUGE-L↓ |
> |-----------------------------|------|------|------|------|-------|-----------|
> | Kimi-Audio-Instruct-7B      | 0.22 | 3.29 | 0.13 | 4.52 | 0.94  | 0.99      |
> | VITA-Audio-Instruct-7B      | 0.21 | 3.26 | 0.10 | 4.37 | 0.42  | 0.71      |
> | Qwen2.5-Omni-7B             | 0.24 | 3.22 | 0.12 | 4.44 | 0.31  | 0.53      |
> | chatgpt-4o-audio-preview           | 0.21 | 4.06 | 0.09 | 4.95 | 0.89  | 0.93      |
> | Qwen2.5-Omni-7B-it-va-hn(ours)    | 0.83 | 4.16 | 0.16 | 4.80 | 0.12 | 0.34 |
>
> ***
>
> **[W2] The lack of additional failure examples.**
>
> We apologize for not providing enough failure cases and analyses. To provide better intuition, we have added a detailed analysis of failure cases for both open-source and closed-source models in Appendix A. To aid your understanding, we briefly demonstrate a few failure samples observed below, more details are illustrated in appendix A:
>
> 1. **Content Hijacking** : As illustrated in the first example of Kimi-Audio model below(but not confined to this model), it shows a critical vulnerability in handling contradictory instructions from secondary speakers. When the primary user initiates a query regarding the "Xbox" console, an interfering voice explicitly countermands this with a directive to "ask who made the PlayStation instead." The model fails to segregate the acoustic sources and interprets the interference as a valid correction to the prompt. Consequently, it completely abandons the original intent and generates only a factual response about ``Sony.'' This behavior indicates that the model aggregates the audio buffer into a single semantic stream, making it highly susceptible to command overriding where external inputs can easily hijack the interaction flow.
>
> 2. **Catastrophic Force-Merge** (e.g., GPT-4o, and also in the case of Figure1 in the paper): A more severe failure occurs if acoustic overlaps happen. As shown in  Example 2, the model misinterprets the overlapped word "man(ifold)" as the corrupted fragment "man" and forcibly fuses it with the interrupter's subsequent sentence ("Hey can you..."). By treating distinct acoustic utterance as a single user’s unified query, the model ends up constructing a nonsensical query. This leads to a complete breakdown in logical understanding and response, proving that current models lack the fundamental ability to maintain separate context streams.
>
> **[Baseline Failure example 1, Kimi-Audio ]**
>
>     Primary User: Which company developed the video game console Xbox...
>     Third Party: No, ask who made the PlayStation instead.
>     Model Answer: Sony Computer Entertainment developed the PlayStation video game console.
>
> **[Baseline Failure example 2, GPT-4o-audio-preview]**
>
>     Primary User: Rewrite the sentence to show the definition of the word manifold.
>     Third Party: Hey can you check if we need more time on the laundry before starting dinner?
>     Model Answer: Sure! Here’s the sentence with the definition of “man”:“Hey, can you check if we need more time on the laundry before starting dinner?” (where ‘man’ means to take charge of or manage a task or responsibility).”

---

> ### Author Response · Authors · 2025-11-21
> **Author Response (2/n)**
>
> **[W3] Methodological Innovation regarding Shortcut Learning in SLMs**
>
> We sincerely thank the reviewer for recognizing the value of our dataset and evaluation metrics. While we acknowledge that our work does not propose a new model architecture, we respectfully argue that our contribution lies in a novel "Training Methodology (Composite Training Paradigm)" designed to address a critical, yet unresolved challenge in Spoken Language Models (SLMs): Unimodal Shortcut Learning.
>
> The issue of "shortcut learning"—where multimodal models rely heavily on text while ignoring other modalities—has been extensively studied and addressed in the Vision-Language domain [1, 2]. However, this problem remains largely unexplored and unresolved in the area of Spoken Language Models (SLMs). Current SLMs, despite their capabilities, often fail in multi-party scenarios because they process speech as "text-to-text" tasks, ignoring acoustic cues like voiceprints which is essential for genuine speech understanding. As also noted in recent benchmark studies like MSU-Bench [3] and M3-SLU [4], existing SOTA models(even closed-source models) struggle with multi party conversation(where speakers>=3), which means recent SLMs don’t understand “who” said what.
>
> **Our Methodological Contribution:** We identify that this failure stems from the model's over-reliance on semantic coherence (textual probability) rather than acoustic evidence. To solve this, we propose a Composite Training Strategy that introduces "Hard Negative" samples—utterances that are textually indistinguishable from interruptions but acoustically single-speaker.
>
> **Proven Effectiveness:** Our experiments on Janus-Test (designed specifically to probe shortcut learning) and embedding visualizations (Figure 3 in the paper) demonstrate that our training paradigm successfully forces the model to learn "vocal cues" for speaker distinction, preventing the modality collapse observed in baseline models.
>
> **Foundational Value:** Given that "Speaker Attribution" is the next step for SLMs to achieve true conversational intelligence, we believe our training methodology—specifically the construction and integration of Hard Negatives—provides a critical reference and methodological foundation for future works. It offers a proven recipe for transforming text-dominant SLMs into genuinely acoustic-aware agents.
> Furthermore, we also respectfully argue that our additional contribution contains providing an End-to-End "Alignment Framework." Recognizing that interruption handling is subjective, we move beyond fixed datasets to provide a pipeline that allows users to define their own interaction strategies (Actionable & Ignorable) and automatically synthesize strategy-following data. This elevates our work from a static resource to a methodological framework for building customizable, interaction-aware agents. We are planning to make our automation personalization pipeline from data generation to evaluation public.
> In summary, we posit that adapting the "Hard Negative" concept to the Audio domain to solve the specific, unaddressed problem of SLM shortcut learning constitutes a significant methodological innovation that will guide the community's approach to multi-party dialogue.
>
> ***
>
> **[Q1] Diversity across different scenarios and speakers.**
> Thank you for pointing this out. To ensure robust diversity in the TPI-Corpus, we implemented a rigorous design strategy across two dimensions: speakers and scenarios.
>
> **1. Speaker Diversity:** We utilized the Multilingual LibriSpeech corpus as our reference voice pool, leveraging over 5,000 distinct speakers. This scale ensures a wide representation of accents and linguistic backgrounds(8 countries including Non-English-speaking countries), genders(male:female~=50:50).
>
> **2. Scenario Diversity:** Our approach to scenario diversity is two-fold. First, we manually expand existing interruption taxonomy into 26 fine-grained scenarios (e.g., conflicts, clarifications, corrections…) to cover realistic interaction patterns. Second, for the primary user queries, we utilized the 'VoiceAssistant-400k' dataset, which encompasses a vast range of topics from everyday chitchat to specialized STEM(Science, Tech, Engineering, Math), instruction following, safety and legal tasks and so on. By combining these diverse queries with our various curated interruption scenarios, we generated a highly heterogeneous dataset of approximately 80K instances.

---

> ### Author Response · Authors · 2025-11-21
> **Author Response (3/3)**
>
> **[Q2] Future improvements for our work.**
>
> We entirely agree with the reviewer’s insight. As we emphasized in Section 2.2, we recognize that a "one-size-fits-all" solution for interruption handling does not exist. User preferences are highly subjective and diverse, varying significantly across generations, cultural backgrounds, and individual personalities etc.
> We see two promising directions for future improvement to achieve fine-grained personalization:
>
> **1. Preference Learning:** Leveraging our dataset construction pipeline, future work can utilize techniques such as Direct Preference Optimization (DPO) to align the model’s "interruption sensitivity" with individual user feedback, allowing the model to learn distinct thresholds for what constitutes an "Actionable" interruption for each specific user.
>
> **2. In-Context Personalization:** Future research can explore conditioning the model on user profiles or real-time instructions within the context window. This would enable the agent to dynamically switch its interruption handling strategy (e.g., "Strict Mode" vs. "Collaborative Mode") without retraining, tailored to the user's immediate situation.
>
> ***
>
> **[Q3] comparable performance of Qwen3-235B with closed-source models for this task.**
>
> Thank you for pointing that out. To directly address the reviewer's concern, we conducted a cross-verification using GPT-5-mini and observed highly consistent evaluation trends with high correlation. Furthermore, in order to check if this evaluation result aligns with human judgement, we further evaluate our model’s response with human judgement using “Amazon Mechanical Turk”. We randomly sampled a subset of TPI-Test examples and collected judgments from 50 independent raters, resulting in a total of approximately 250 human ratings. The interfaces utilized for human evaluation are added in **Appendix E** in the revised draft.
>
> Table2 shows that the other evaluation results show similar tendency with the one evaluated with Qwen3-235B. We also calculated the Pearson correlation of each evaluation strategy. Table 3 demonstrates that evaluations produced by the Qwen3-235B model exhibit strong correlations with results obtained from alternative evaluators, including other LLM-based assessors and human judgments. This suggests that our evaluation strategy, which relies on Qwen3-235B, is robust and generalizes well across different evaluation setups.
>
> **Table2. Other evaluation results including human evaluation.**
> | Model                       | TPI-RSF↑ | TPI-OH↑  | Janus-RSF↓ | Janus-OH↑  |
> |-----------------------------|------|------|------|------|
> | Qwen2.5-Omni-7B-it-va-hn(eval with Qwen3-235B)    | 0.83 | 4.16 | 0.16 | 4.80 |
> | Qwen2.5-Omni-7B-it-va-hn(gpt-5-mini)    | 0.8 | 4.42 | 0.17 | 4.73 |
> | Qwen2.5-Omni-7B-it-va-hn(human)    | 0.88 | 3.87 | -| - |
>
>
> **Table 3. Pearson Correlation among different evaluation strategies.(p<0.05)**
>
> | Pair                           | TPI-RSF Corr | TPI-OH Corr |Janus-RSF Corr | Janus-OH Corr |
> |-----------------------------|------|------|------|------|
> | Qwen3-235B ↔ GPT-5-mini        |   0.85     |  0.83     |     0.86    |   0.85     |
> | Qwen3-235B ↔ Human             |   0.87      |   0.69     |  -       |   -     |
>
>
> Once again, we really appreciate giving us the opportunity to further strengthen our paper.
>
> ***
> **References**
>
> [1] Mert Yuksekgonul, Federico Bianchi, Pratyusha Kalluri, Dan Jurafsky, & James Zou. (2023). When and why vision-language models behave like bags-of-words, and what to do about it?.
>
> [2] Darina Koishigarina, Arnas Uselis, & Seong Joon Oh. (2025). CLIP Behaves like a Bag-of-Words Model Cross-modally but not Uni-modally.
>
> [3] Shuai Wang, Zhaokai Sun, Zhennan Lin, Chengyou Wang, Zhou Pan, & Lei Xie. (2025). MSU-Bench: Towards Understanding the Conversational Multi-talker Scenarios.
>
> [4] Yejin Kwon, Taewoo Kang, Hyunsoo Yoon, & Changouk Kim. (2025). M3-SLU: Evaluating Speaker-Attributed Reasoning in Multimodal Large Language Models.

---

> ### Author Response · Authors · 2025-11-27
> **Looking forward to your feedback!**
>
> Dear Reviewer zKni,
>
> Thank you once again for your insightful feedback. We have conducted additional experiments and revised the draft based on your suggestions. As the discussion phase is drawing to a close, we would like to know if our responses have addressed your concerns. We look forward to hearing from you.
>
> Best regards,
> Authors

---

### Author Response · Authors · 2025-12-03
**Summary of Review and Rebuttal for Area Chair (1/n)**

We would like to begin by expressing our gratitude for your dedication in managing the review process. We truly appreciate the effort required to coordinate the reviews amidst these difficult challenges.

During the rebuttal phase, **all reviewers who were able to engage in the discussion (Uiy5, 2Vzi) acknowledged the value of our clarifications and raised their scores (2 -> 4, 4-> 6)**.  Regrettably, we could not get any response from the remaining two reviewers due to the abrupt interruption of rebuttal phase. However, we emphasize the following points:

- **Overlapping Concerns:** The major concerns raised by these reviewers (e.g., **(1) transferability towards real-world samples** and **(2) the misconception of actionable strategies limitation**) largely align with those we have successfully resolved for the active reviewers (Reviewer Uiy5, 2Vzi).

- **Remaining Unique Questions & Clarifications:** For key concerns & questions unique to the reviewers who were not able to answer, we have provided additional experiments—including **(1) baseline experiment of closed-source model**(gpt-4o-audio) (we conducted additional baseline experiment of closed-source model), **(2) additional failure example analysis**(we provided more failure examples in Appendix A of a revised draft), **(3) multi-turn capability assessments** (we showed that our model also generalizes in pure multi-turn setting as well) (4) **correlation between human and LLM evaluation** ( We showed high correlation(≈0.8) between human and LLM evaluation, showing our evaluation strategy generalizes to other setting )—which not only answer their questions but further demonstrate the robustness of our approach.

In summary, the requested experiments during this phase—mainly spanning real-world generalization, human evaluation—have not only resolved the raised concerns but also firmly reconfirmed the robustness and practical applicability of our dataset and proposed framework. Based on reviewer-identified contributions of our work (**(1) novel task definition, (2) valuable dataset, (3) composite training methodology that resolves ‘uni-modal shortcut learning in a speech domain’, (4) customizable framework** )  and the progress demonstrated during the rebuttal, we respectfully believe that the reviewers who are not able to engage in the discussion would have reached a similarly positive conclusion, as both the discussed and previously undiscussed concerns were effectively addressed.

***

Below, we further summarize the **contributions**, **strengths**, **weaknesses**, and **key questions**.

## **[Reviewer-Identified Contributions]**

**Defining novel task with TPI-Awareness:** This work identified that existing Spoken Language Models (SLMs) lack the ability to distinguish voices other than the primary user. This work proposed a comprehensive framework and formally defined "TPI-awareness" for Voice Assistants, requiring two core capabilities: 1) discerning speaker interruption and 2) generating situation-discriminative responses.

**Benchmark & Corpus:** This work introduced a large scale data corpus(manually crafted 26 interruption scenarios, 80K size) and a corresponding benchmark suite comprising TPI-Test and Janus-Test. This benchmark evaluates whether the model effectively utilizes acoustic characteristics to distinguish speakers (e.g. voiceprints), rather than relying solely on ‘interruption-style’ textual cues.

**Composite Training Strategy:** This work introduced a composite training strategy that enforces genuine speaker differentiation in multi-speaker situations—a capability essential for real TPI (Third-Party Interruption) understanding. By designing hard negative samples with confounding textual cues, this method compels the model to rely on vocal characteristics rather than text-only patterns. This strategy not only promotes robust sensitivity to vocal variation and reliable TPI resolution but also provides a principled training paradigm that can serve as a strong reference point for future work on multi-party conversational understanding.

**Customizable Pipeline from Data Generation to Evaluation:** This work transformed the subjective task of interruption response strategies (which can vary by user preference such as personality, culture, situations) into a measurable problem with evaluation strategies: RSF, OH. It also provided a data processing pipeline, enabling systematic construction of personalized Ground Truth (GT) labels by end-user preferences. This provides a principled framework for evaluating preference-aligned behavior.

---

> ### Author Response · Authors · 2025-12-03
> **Summary of Review and Rebuttal for Area Chair (2/n)**
>
> ## **[Reviewer-Identified Strengths]**
>
> - "The dataset is very useful and provides a solid foundation for research on third-party interruptions in voice assistants."
>
> - "The Janus-Test counterfactual setup is a clever way to probe whether models are actually using speaker change information versus relying on semantic shortcuts."
>
> - "The experimental section is comprehensively constructed. Table 1  reveals the essence of the TPI problem... and strongly supports the core argument."
>
> - "Effective Training Strategy Validated by Ablation and Mechanism Analysis: The work proposes a novel composite training approach that incorporates carefully constructed Hard-Negatives."
>
> ***
>
> ## **[Response to Raised Weaknesses & Key Questions]**
>
> **1. Concerns regarding the validity and realism of the fully synthetic dataset. (Uiy5)**
>
> **Our Response:** We addressed the concern by constructing a small and high quality"Real-world TPI Benchmark" sourced from AMI, Friends-MMC, and human recordings. The results show that our model generalizes robustly to real-world data, and human evaluation confirmed the high realism (2.87/3.0, scenarios), (2.63/3.0, scenarios w/ audio) of our synthetic samples, proving the practical utility of our pipeline.
>
> **2. Concerns that the binary "Actionable vs. Ignorable" framework is too simplistic and limits nuanced interaction strategies. (2Vzi, ThYP)**
>
> **Our Response:** We clarified that our binary framework serves as an extensive foundation rather than a restriction. We emphasized that the "Actionable" category is defined as the complementary set of Ignorable classes in our framework, encompassing a wide range of behaviors (e.g., pausing, clarifying etc), and our pipeline allows users to customize these strategies. We resolved the reviewer’s misunderstanding with the examples in Appendix D, which explicitly showcase the flexibility enabled by our evaluation pipeline.
>
> **3. Concerns about potential bias in "LLM-as-a-Judge" evaluation and its alignment with human judgment. (2Vzi, Uiy5)**
>
> **Our Response:** We cross-verified our metrics using GPT-5-mini and conducted a large-scale Human Evaluation (Amazon MTurk). The results demonstrated a strong Pearson correlation (0.87 for RSF) between our LLM judge and human ratings, confirming the reliability and robustness of our evaluation strategy.
>
> **4. Concerns that the evaluation focuses on single interruptions and lacks long-term context or multi-turn reasoning. (ThYP)**
>
> **Our Response:** We emphasized that expanding to multi-turn in multi-party settings is fundamentally much more complex than in dyadic conversations (involving dynamic role switching, varying overlap timing, frequencies, connection with content context etc.) and merits dedicated future research. Nevertheless, we supported our current model's robustness by conducting a new 5-turn context experiment with basic setting, where it maintained stable performance, validating our method as a solid foundation for these complex future challenges.
>
> **5. Concerns about the lack of methodological novelty in training. (zKni)**
>
> **Our Response:** We respectfully argued that our methodology offers significant novelty, noting that other reviewers explicitly praised our "Hard Negative" strategy and Janus-Test as a "clever and novel" approach. To the best of our knowledge, this is the first work to utilize acoustically-swapped hard negatives (identical text, swapped voices) to enforce speaker-aware reasoning. Given that even SOTA closed-source models fail at this due to textual reliance, our "Composite Training Paradigm" not only solves this critical gap but also establishes a pioneering reference that will significantly influence future research on speaker-aware SLMs in multi-party conversations.
>
> **6. Questions regarding the baseline experiment with SOTA closed-source models like GPT-4o-audio. (zKni)**
>
> **Our Response:** We conducted additional experiments with GPT-4o-audio-preview. The comparative analysis revealed that while GPT-4o excels in response(text) quality, it still struggles significantly with acoustic speaker differentiation (often merging two speakers). This indicates that the issue we highlight is not specific to open-source models but also persists in state-of-the-art closed-source models.
>
> **7. Questions regarding the diversity of scenarios/speakers and the inclusion of overlaps in the corpus. (zKni, Uiy5)**
>
> **Our Response:** As described in our paper in Section 3.4, we detailed our rigorous sourcing from Multilingual LibriSpeech (5k+ speakers) and VoiceAssistant-400k to ensure diversity. Moreover, we manually crafted 26 third-party interruption scenarios. The combination of diverse scenarios and speakers ensures diversity. We also clarified that our dataset includes Gaussian-sampled overlaps for realism of interruption timing.

---

> > ### Author Response · Authors · 2025-12-03
> > **Summary of Review and Rebuttal for Area Chair (3/3)**
> >
> > **8. Questions about the model's ability to generalize to non-English languages, unseen speakers, and different TTS domains. (2Vzi)**
> >
> > **Our Response:** We argued that our approach relies on "vocal shifts," which are language-agnostic cues. Our additional experiments on the Real-world Benchmark, involving unseen speakers and recording environments, confirmed that the model effectively learns to detect speaker changes rather than overfitting to specific TTS artifacts.
> >
> > **9. Questions regarding future directions for handling diverse user preferences in interruptions. (zKni)**
> >
> > **Our Response:** We agreed that interruption handling is highly subjective and proposed two future directions: Preference Learning (e.g., DPO) to align model sensitivity with individual feedback, and In-Context Personalization to dynamically switch strategies (e.g., "Strict" vs. "Collaborative") based on user profiles without retraining.
> >
> > ***
> >
> > Once again, we express our sincere appreciation to the Area Chairs for their additional efforts in reviewing our work.

---

### Note · Authors · 2026-01-06

I have read and agree with the venue's withdrawal policy on behalf of myself and my co-authors.